# Working conditions of the clinical health workforce in the public health facilities in Bangladesh

**Syed Abdul Hamid**[1,2⊛], **Md. Ragaul Azim**[1⊛]*, **Md. Mahfujur Rahman**[1], **Md. Sirajul Islam**[2]

**1** Institute of Health Economics, University of Dhaka, Dhaka, Bangladesh, **2** Universal Research Care Ltd., Dhaka, Bangladesh

⊛ These authors contributed equally to this work.
* azim.ihe@du.ac.bd

**Data Availability Statement:** All relevant data are within the paper and its Supporting Information files.

## Abstract

### Background

The COVID-19 pandemic has highlighted the importance of a well-equipped and supported healthcare workforce, and Bangladesh still faces challenges in providing adequate and well-equipped healthcare services. Therefore, the study aims to assess the level of working conditions of the clinical health workers in Bangladesh and their relative importance in delivering quality healthcare services.

### Methods

The study followed a cross-sectional study design and collected primary data adopting a quantitative method. A total of 319 clinical workforces from four districts and eight sub-districts were randomly selected using a multi-stage sampling technique. A 26-component questionnaire used to assess various components of working conditions. Descriptive statistics, and bivariate analysis were used to analyze the data.

### Results

The study found that the working conditions of clinical health workers in primary and secondary healthcare facilities in Bangladesh were quite poor (3.40), with almost two-thirds of respondents showing negative views in 23 out of 26 indicators. The results also showed that working conditions were significantly ($p \leq 0.05$) higher in primary compared to secondary level facilities. Moreover, men, younger workforce, and workforce with shorter length of service were more likely to report poor working conditions than their counterparts. Lastly, receiving monthly salary in due time was top-ranked (99.15) in terms of importance for delivering quality healthcare, followed by availability of medicines (98.04), and medical and surgical requisites (97.57), and adequate mentoring and support to perform duties (97.50).

**Funding:** SAH received fund for this research work. This study was founded by Centennial Research Grant (CRG), University of Dhaka, Bangladesh. The funder website URL is as follows: https://www.du.ac.bd/. But the funder did not provide any grant number for this grant. The funder had no role in study design, data collection and analysis, decision to publish, or preparation of the manuscript.

**Competing interests:** The authors have declared that no competing interests exist.

## Conclusion

The study highlights the poor working conditions of clinical health workers in public health facilities in Bangladesh. It recommends that policymakers should prioritize improving working conditions by addressing the factors that are crucial for delivering quality healthcare. Improving working conditions will have a positive impact on the retention and motivation of workers, which will ultimately lead to better health outcomes for the population.

## 1. Introduction

Bangladesh has made remarkable strides in improving access to healthcare for its citizens in recent years, but challenges still remain in ensuring a sufficient and well-equipped health workforce to deliver quality healthcare. The COVID-19 pandemic has further emphasized the importance of a well-equipped and supported healthcare workforce in responding to health crises. One critical aspect of the health workforce is their working conditions, which significantly affects their job satisfaction, retention, and ultimately the quality of care [1–3]. The government of Bangladesh has adopted various strategies to improve working conditions in healthcare facilities including construction of new healthcare facilities and the renovation of existing ones, and recruiting and training more healthcare workers [4]. Still, health workers in Bangladesh have reported a range of challenges in their working conditions, including low salaries, inadequate equipment and supplies, long working hours, and lack of supportive supervision and training [1]. These factors can lead to burnout, stress, and poor health outcomes, and can contribute to health worker migration to other countries or to the private sector; exacerbating the shortage of health workers in the public sector [3].

Decent working conditions of the health workforce are paramount to delivering quality healthcare services at an affordable cost. Several studies claim an association between quality of care and individual factors of working conditions [5, 6]. Working conditions have been argued as the most crucial factor for the satisfaction of the health workforce that led to the quality of health care services [7]. As evidenced, individual components of working condition have a significant impact on the quality, effectiveness, and efficiency of work and also on healthcare costs. A growing body of literature also claims the relationship between working conditions and the satisfaction of a health workforce. Study shows that satisfied workers are more productive and committed to work, whereas dissatisfied ones tend to practice absenteeism, grievances, and ineffective and inefficient behavior [8, 9]. Some studies show that the satisfaction of health workers is correlated with the work environment, incentive, job recognition, promotion opportunities, workloads, and relation of employees with their supervisors [10, 11]. Therefore, satisfaction and motivation of the health workers are the crucial factors to provide quality healthcare services for achieving Sustainable Development Goals (SDGs) [7, 12, 13]. The positive association of working conditions and job satisfaction is supported by the job demands-resources (JD-R) model which provides a useful theoretical and conceptual framework for understanding the factors that contribute to the workload and well-being of healthcare workers. Job demands include factors such as workload, time pressure, emotional demands, and role ambiguity, while job resources include social support, autonomy, feedback, and opportunities for learning and development [14, 15]. The JD-R model predicts that high job demands and low job resources can lead to negative outcomes such as burnout and turnover intentions, while high job resources can lead to positive outcomes such as work engagement and job satisfaction [14].

Work commitment is vital, especially for the healthcare workforce. Studies show that different components of working condition play an important role to determine employees' commitment to work which is associated with the provision, quality, accessibility, and affordability of healthcare [8]. Thus, working conditions affect the efficiency and effectiveness of the healthcare team. Health workers who work in an environment with inadequate resources and equipment may face delays in providing care, which can compromise patient outcomes [16]. Adequate staffing levels, equipment, and supplies are necessary for efficient healthcare delivery [17]. Moreover, workload and job stress can affect the efficiency of health workers, leading to burnout and turnover [18]. Therefore, upholding good working conditions is necessary to deliver quality healthcare services at an affordable cost to achieve universal health coverage (UHC).

There is limited research on measuring the status of working conditions of clinical health workers in both national and international context. As stated above, some studies have focused on specific aspects of health workers' working conditions in Bangladesh, such as job satisfaction or motivation. However, a comprehensive analysis of the various dimensions of health workers' working conditions is lacking. To fill up this knowledge gap, the current study examined the level of various components of working conditions, including their workload, remuneration, training, workplace layout, safety, water supply, job recognition, and other support systems. This study also assessed the relative importance of the different factors of working conditions to deliver quality care. The findings of the study will be of interest to policymakers, healthcare managers, and researchers, both in Bangladesh and globally. Ultimately, this study could contribute to the development of evidence-based policies and practices to improve the working conditions of healthcare workforce.

## 2. Subjects and methods

### 2.1 Sampling methods

The study followed a cross-sectional study design and analyzed the primary data collected from January to March 2022 applying a quantitative method. We adopted a multi-stage sampling technique to select a representative sample of the clinical health workforce. Four districts out of 64 and eight upazilas (the second lower tier of the administrative structure) out of 495 from the four greater administrative divisions (Dhaka, Chattogram, Rajshahi, and Khulna) of Bangladesh were selected as the study area. In the first stage, the study randomly selected four districts drawing one from each of the four administrative divisions. In the second stage, a total of eight upazilas, taking two from each of the selected districts were selected using the same technique. In the third stage, the study employed a random sampling technique to select clinical workforce from the both District Hospital (DH) and Upazila Health Complex (UHC) of the selected districts and upazilas. The clinical workforce included physicians such as Medical Officer (MO), Emergency Medical Officer (EMO), Junior Consultant, and Senior Consultant, and other clinical staff such as Nurse, Midwife, Medical Technologist, Sub-Assistant Community Medical Officer (SACMO), and Pharmacist. To ensure the representativeness of the sample, the study utilized a systematic sampling method, which involved using a list of all clinical staff in the selected healthcare facilities for the randomization of the respondents. At the outset, we contacted 350 clinical workforces for interviews, consisting of 120 physicians and 230 other clinical staff, to serve as an indicative sample. Regrettably, 31 of them declined to participate. Consequently, we conducted interviews with 319 clinical workforces, comprising 109 physicians and 210 other clinical staff. This sample size determination was made considering various factors, including the time, budget, and practical feasibility of conducting interviews with clinical workforces in the context of Bangladesh. Physicians and other clinical staff often have demanding schedules and limited availability for research interviews.

Additionally, public health facilities often experience a high flow of patients, further reducing the time healthcare providers can allocate for interviews. Given these practical limitations, we opted for an indicative sample size that was feasible to conduct within the available resources and timeframe. The multi-stage sampling ensured the representativeness of the sample of the people of interest, and allowed for the collection of data from a diverse range of geographic and administrative areas.

## 2.2 Data collection instruments

For conducting the survey, we used a structured questionnaire containing both socio-demographic characteristics of the respondents and different indicators of working conditions. Working conditions have been defined differently in the literature. One study explained working conditions as working environment and other factors influencing the motivation of the workforce to perform their duties [19]. The other studies defined working conditions as the environment where physical and psychological factors affect one's work [20, 21]. Consulting with relevant literature, we prepared the questionnaire comprising twenty-six (26) components of working conditions, including workload, workplace layout, safety, water supply, and job recognition [20, 22, 23]. The questionnaire contained separate statements for each component of working conditions for examining the perceived status of the working conditions of the clinical workforces as stated in Table 1.

During the initial phase of the study, we conducted a pilot testing of the questionnaire to assess its reliability and validity within the country context. The pilot involved a sample of 30 clinical providers from various healthcare facilities resembling our study population. The primary objective of the pilot was to evaluate the clarity, coherence, and appropriateness of the questionnaire items for our target respondents. We utilized the feedback received from the pilot participants to refine the questionnaire's design. Finally, we translated the questionnaire into Bengali, the local language spoken by the study participants, and administered by the trained enumerators to ensure its accessibility and cultural appropriateness. Each statement evaluated using 5 points Likert scale (1 = strongly disagree, 2 = disagree, 3 = neutral, 4 = agree, and 5 = strongly agree). The study received ethical approval from the Institutional Review Board (IRB) of the Institute of Health Economics, University of Dhaka. The study obtained verbal informed consent from all participants. We explained the purpose and nature of the study to each participant, including the voluntary nature of their participation, the confidentiality of their responses, and their right to withdraw from the study at any time without consequences. The research team documented the verbal consent, and notarized by an impartial witness present during the consent process.

## 2.3 Data analysis

We utilized both descriptive statistics and bivariate analysis to analyze the data. Specifically, the study calculated the mean score of all the statements regarding each component of the working conditions. Additionally, the study reported the percentage of respondents who agreed or strongly agreed with the statements related to good working conditions. For poor working conditions, the study merged the neutral option with the 'disagree', and 'strongly disagree' options. This was done to account for respondents who may have selected the neutral option due to indifference or lack of positivity toward the statement in question. To determine the overall status of the working conditions, the study computed a weighted mean score that accounted for the weight given by the clinical workforces to each component, as well as their reported score for that component. The purpose of computing the weighted mean was to account for the varying significance that clinical workforces might assign to different

**Table 1. List of statements including abbreviated form used to assess working conditions.**

| Serial | Statements used to assess working conditions in 5-point Likert scale (1 = strongly disagree, 2 = disagree, 3 = neutral, 4 = agree, and 5 = strongly agree) | Short-form of the statements used in the result tables |
|---|---|---|
| 1 | The volume of work assigned to me is justifiable | Work volume |
| 2 | My workplace has enough staff to provide quality care | Enough staff |
| 3 | My job roles and responsibilities are clearly defined to perform my assigned duties without interruptions and distractions | Job roles and responsibilities |
| 4 | I have received the required training to perform my assigned duties | Required training received |
| 5 | My current rank is matched with the duration of my job experience | Compatibility of rank |
| 6 | The layout of my working place does not hamper my work efficiency | Workplace layout |
| 7 | There is the adequacy of functional medical equipment to perform my assigned duties efficiently | Functional medical equipment |
| 8 | There are proper safety measures against violence in my workplace | Provider's safety |
| 9 | I do not often feel stress in my workplace | Stress-free workplace |
| 10 | I do not have major deprivations (e.g., promotion missed, undue failure in post-graduation education, etc.) in my current job | Burnout |
| 11 | I get proper recognition for my work | Recognition of work |
| 12 | I have a good relationship with my co-workers | Relationship with co-workers |
| 13 | My workplace is adequately clean and odor free | Clean workplace |
| 14 | Color and lighting condition is compatible with my work | Color and lighting condition |
| 15 | My workplace is well ventilated | Ventilation |
| 16 | My workplace is designed with proper aesthetic views | Aesthetic workplace |
| 17 | Washrooms are adequate and usable | Washroom facility |
| 18 | My workplace is noise and sounds free | Noise and sound-free workplace |
| 19 | My workplace has an uninterrupted power supply | Uninterrupted power supply |
| 20 | My workplace has an uninterrupted water supply | Uninterrupted water supply |
| 21 | My workplace is damp free | Damp-free workplace |
| 22 | I find that my opinions are respected in the workplace | Respect to opinions |
| 23 | The manager supports and values my work | Valuation of work |
| 24 | I find adequate mentoring and support to assist in my duties | Mentoring |
| 25 | I receive my monthly salary in due time | On-time salary |
| 26 | My workplace has consistent availability of medicines and Medical and Surgical Requisites (MSR) to perform my duties | Medicine and MSR supply |

components when assessing their working conditions. This approach allowed us to give appropriate weightage to the responses, reflecting their relative importance in contributing to the overall status of working conditions. We aimed to provide a more accurate and comprehensive evaluation by considering both the respondents' scores for each component and their individual significance. We used the 75th percentile as a cut-off point to categorize the working conditions as either good or poor, based on our understanding of the issue and the study's context [24]. The 75th percentile cut-off point was used to distinguish between higher and lower assessments of working conditions among the study participants. It represents a threshold that separates the upper quartile of responses from the lower three quartiles. By applying the 75th percentile cut-off, we aimed to identify and categorize respondents who reported relatively more positive assessments of working conditions as "good" conditions, while those who reported relatively lower assessments were categorized as "poor" conditions.

We also calculated the mean weight score of the different components of working conditions to assess and rank their relative importance to delivering quality healthcare. In order to calculate the mean weight score, we assigned a 100-point scale (weight) to each component.

The respondents were asked to rank the perceived importance of these components in delivering quality healthcare, with higher scores indicating greater importance. We obtained the individual weight scores assigned by each respondent for a particular component and then calculated the average of these scores across all respondents. Finally, the study used non-parametric tests to assess the significance of differences between the responses of different categories of respondents. Specifically, the study utilized the Wilcoxon rank-sum test to compare the responses of two groups and the Kruskal-Wallis test to compare the responses of three or more groups.

Lastly, we conducted a psychometric analysis to assess the construct validity, and reliability of the survey tools utilized in the study. The construct validity of the survey instrument was tested using two measures, namely factor analysis and Cronbach's alpha. Confirmatory factor analysis (CFA) was used to test whether all the indicators used in the survey can explain a single latent construct called working conditions though the small sample size might pose constraints on the stability of factor loadings. The factor loadings estimated from CFA measured the correlation between responses to the questions and the unobserved variable called construct (i.e., working conditions). The cutoff point of factor loadings differs across studies with some literature considering a significant factor loading as $> = 0.4$, while others require $>0.5$ [25–27]. To assess the internal consistency reliability, we estimated the Cronbach alpha coefficient which ranges from 0 to 1 and is a measure of the unidimensionality of the indicators. The cutoff value of alpha coefficients ranges from 0.6 to 0.8 based on evidence from prior study [28].

## 3. Results

The study interviewed a total of 319 clinical health workforce, with 45 physicians and 123 other clinical staff from the UHCs, and 64 physicians and 87 other clinical staff from the DHs. The age of the respondents ranged from 23 to 59 years. More than half of them were below 40 years of age. About 58% of the respondents were female, and 42% were male (Table 2). A large portion (41.4%) of the respondents were nurses and midwives, followed by MO, EMO, and RMO (27.6%), medical technologist (13.8%), consultant (6.6%), SACMO (5.3%), and pharmacist (5.3%). More than one-third of the respondents had 11 to 39 years of experience in public hospitals (Table 2).

The working conditions of the clinical health workforce in the primary and secondary level healthcare facilities in Bangladesh were found quite poor, as indicated by the estimated weighted mean score of 3.4 (Table 3). We found that 23 out of 26 indicators were reported negatively by the respondents. Specifically, a significant proportion of respondents expressed negative views regarding factors such as sufficient staffing, workplace layout not affecting work efficiency, inadequate safety measures against violence, and frequent experience of workplace stress. However, more than two-thirds of respondents showed a positive view regarding uninterrupted water supply, managerial support and recognition, mentoring and support, and consistent availability of medicines and Medical and Surgical Requisites (MSR) to perform their duties. Furthermore, over 90% of the respondents indicated a good relationship with their co-workers and timely receipt of monthly salary (Table 3).

The analysis revealed significant differences in the perception of good working conditions between UHC and DH facilities in approximately half of the indicators ($p \le .05$). UHC facilities demonstrated a significantly higher likelihood of reporting good working conditions compared to DH facilities in 11 out of 26 indicators. Conversely, DH facilities reported significantly higher good working conditions than UHC facilities in only two indicators: timely receipt of monthly salary and consistent availability of medicines and MSR (Table 4). Furthermore, significant differences ($p-\le .05$) existed in reporting good working conditions

**Table 2. Socio-demographic characteristics of the respondents.**

| Attributes | | | n | % |
|---|---|---|---|---|
| Gender | Male | | 134 | 42 |
| | Female | | 185 | 58 |
| Age | <40 | | 194 | 60.8 |
| | ≥40 | | 125 | 39.2 |
| Workforce type | Physician | | 109 | 34.2 |
| | Other clinical staff | | 210 | 65.8 |
| Workforce category | MO, EMO, and RMO | | 88 | 27.6 |
| | Consultant | | 21 | 6.6 |
| | Nurse and Midwife | | 132 | 41.4 |
| | SACMO | | 17 | 5.3 |
| | Medical Technologist | | 44 | 13.8 |
| | Pharmacist | | 17 | 5.3 |
| Years of service in the public hospital | 1–5 Years | | 145 | 45.5 |
| | 6–10 Years | | 52 | 16.3 |
| | Above 10 Years | | 122 | 38.2 |
| Educational qualification | MBBS | | 77 | 24.1 |
| | Postgraduate medical degree | | 30 | 9.4 |
| | BSc/Diploma in nursing | | 111 | 34.8 |
| | BSc/ Diploma in medical technology | | 42 | 13.2 |
| | Others | | 59 | 18.5 |

Note:
• MO: Medical Officer
• EMO: Emergency Medical Officer
• RMO: Residential Medical Officer
• SACMO: Sub-Assistant Community Medical Officer

between physicians and other clinical staff in 18 out of 26 indicators. Other clinical staff members were more likely to report good working conditions compared to physicians in all indicators (where significant differences exist), except for the alignment of current rank with job experience duration.

Table 5 revealed that men exhibit a higher likelihood of reporting poor working conditions compared to women in 20 out of 26 indicators. Significant differences (p≤.05) existed in 10 of these indicators. Conversely, women reported significantly higher poor working conditions than men for the indicator related to receiving required training (p≤.05). Additionally, respondents aged 40 years or above (older) were more likely to report good working conditions compared to those below 40 years old (younger) in 23 out of 26 indicators. Significant differences (p≤.05) were found in 9 of these indicators. However, older individuals experienced significantly higher poor working conditions than younger counterparts for two indicators: alignment of current rank with job experience and absence of major deprivations in the current job (e.g., missed promotions, undue failures in post-graduation education). Similarly, individuals with a long duration of service (over ten years) in public hospitals were more likely to report good working conditions compared to those with a shorter duration of service in almost all indicators. Significant differences (p≤.05) were observed in 8 indicators. Like the older group, long-serving individuals experienced significantly higher poor working conditions compared to their counterparts for the two aforementioned indicators (p≤.05).

**Table 3. Status of working conditions of the health workforce in the primary and secondary level health facilities, and mean weight score of the indicators.**

| Indicators | Good working conditions | | Poor working conditions | | Mean Score on a scale of 5 | Mean weight score on a scale of 100 | Rank based on importance (mean weight score) |
|---|---|---|---|---|---|---|---|
| | % | n | % | n | | | |
| Work volume | 56.4 | 180 | 43.6 | 139 | 3.4 | 95.33 | 22nd |
| Enough staff | 27.6 | 88 | 72.4 | 231 | 2.4 | 96.00 | 16th |
| Job roles and responsibilities | 60.5 | 193 | 39.5 | 126 | 3.6 | 95.98 | 17th |
| Required training received | 39.2 | 125 | 60.8 | 194 | 2.9 | 95.85 | 19th |
| Compatibility of rank | 55.5 | 177 | 44.5 | 142 | 3.3 | 96.71 | 10th |
| Workplace layout | 35.7 | 114 | 64.3 | 205 | 2.9 | 93.71 | 24th |
| Functional medical equipment | 46.7 | 149 | 53.3 | 170 | 3.1 | 96.93 | 8th |
| Provider's safety | 32.0 | 102 | 68.0 | 217 | 2.6 | 96.15 | 14th |
| Stress-free workplace | 31.0 | 99 | 69.0 | 220 | 2.7 | 95.77 | 20th |
| Burnout | 37.6 | 120 | 62.4 | 199 | 2.7 | 96.34 | 12th |
| Recognition of work | 49.8 | 159 | 50.2 | 160 | 3.3 | 95.97 | 18th |
| Relationship with co-workers | 96.2 | 307 | 3.8 | 12 | 4.5 | 97.50 | 4th |
| Clean workplace | 58.6 | 187 | 41.4 | 132 | 3.5 | 96.77 | 9th |
| Color and lighting condition | 60.5 | 193 | 39.5 | 126 | 3.6 | 94.09 | 23rd |
| Ventilation | 65.5 | 209 | 34.5 | 110 | 3.7 | 96.03 | 15th |
| Aesthetic workplace | 40.1 | 128 | 59.9 | 191 | 3 | 91.29 | 25th |
| Washroom facility | 41.4 | 132 | 58.6 | 187 | 2.9 | 96.25 | 13th |
| Noise and sound-free workplace | 57.4 | 183 | 42.6 | 136 | 3.4 | 95.74 | 21st |
| Uninterrupted power supply | 56.6 | 180 | 43.6 | 139 | 3.4 | 97.41 | 5th |
| Uninterrupted water supply | 73.4 | 234 | 26.6 | 85 | 3.9 | 97.28 | 6th |
| Damp-free workplace | 63.6 | 203 | 36.4 | 116 | 3.6 | 96.45 | 11th |
| Respect to opinions | 61.1 | 195 | 38.9 | 124 | 3.6 | 96.45 | 11th |
| Valuation of work | 67.7 | 216 | 32.3 | 103 | 3.8 | 97.13 | 7th |
| Mentoring | 75.9 | 242 | 24.1 | 77 | 3.9 | 97.57 | 3rd |
| On-time salary | 94.4 | 301 | 5.6 | 18 | 4.7 | 99.15 | 1st |
| Medicine and MSR supply | 71.2 | 227 | 28.8 | 92 | 3.8 | 98.04 | 2nd |
| The calculated weighted mean score | 3.40 | | | | | | |

Note:

• MSR: Medical and Surgical Requisites

The reported good working conditions significantly varied among various groups of the health workforce in 20 out of 26 indicators. There is no specific category of health workforce reporting consistently higher or lower overall good working conditions; rather, it contrasted across indicators. Over 90% of workforces from all categories reported having a good relationship with their co-workers and receiving their monthly salary on time. Approximately 70% of consultants stated that they have received the necessary training for their assigned duties and that their current rank aligns with their job experience duration. In contrast, more than two-thirds of pharmacists expressed the opposite situation. More than 90% of consultants indicated a lack of proper safety measures against workplace violence, inadequate recognition of work, insufficient staffing, and a lack of functional medical equipment to perform their duties, which was also echoed by over 70% of medical officers (MO, EMO, RMO) (Table 5). There was a significant difference ($p \leq .05$) in reporting good working conditions among workforces with different educational levels in 20 indicators. Medical officers and consultants were more likely to report poor working conditions than other clinical staff for all indicators except 'receiving required training' and 'matching current rank with job experience duration.'

**Table 4. Percentage rating working conditions as good (represented by percent of respondents who rated the indicators as 'agree' and 'strongly agree') for different indicators by the type of health facilities, and types of workforces.**

| Indicators | Type of health facility | | | Type of health workforce | | |
|---|---|---|---|---|---|---|
| | Primary Healthcare Facility (UHC) | Secondary Healthcare Facility (DH) | P-value | Physician | Other clinical staff | P-value |
| | n (%) | n (%) | | n (%) | n (%) | |
| Work volume | 110 (65.5) | 70 (46.4) | 0.00 | 53 (48.6) | 127 (60.5) | 0.43 |
| Enough staff | 56 (33.3) | 32 (21.2) | 0.01 | 10 (9.2) | 78 (37.1) | 0.00 |
| Job roles and responsibilities | 112 (66.8) | 81 (53.6) | 0.02 | 52 (47.7) | 141 (67.1) | 0.00 |
| Required training received | 68 (40.5) | 57 (37.8) | 0.62 | 49 (45.0) | 76 (36.2) | 0.13 |
| Compatibility of rank | 89 (53.0) | 88 (58.3) | 0.34 | 73 (67.0) | 104 (49.5) | 0.00 |
| Workplace layout | 60 (35.7) | 54 (35.8) | 0.99 | 37 (33.9) | 77 (36.7) | 0.63 |
| Functional medical equipment | 78 (46.4) | 71 (47.0) | 0.92 | 28 (25.7) | 121 (57.6) | 0.00 |
| Provider's safety | 56 (33.3) | 46 (30.5) | 0.58 | 12 (11.0) | 90 (42.9) | 0.00 |
| Stress-free workplace | 61 (36.3) | 38 (25.2) | 0.03 | 19 (17.4) | 80 (38.1) | 0.00 |
| Burnout | 68 (40.5) | 52 (34.4) | 0.27 | 44 (40.4) | 76 (36.2) | 0.47 |
| Recognition of work | 93 (55.4) | 66 (43.7) | 0.04 | 34 (31.2) | 125 (59.5) | 0.00 |
| Relationship with co-workers | 165 (98.2) | 142 (94.0) | 0.05 | 104 (95.4) | 203 (96.7) | 0.58 |
| Clean workplace | 106 (63.1) | 81 (53.6) | 0.09 | 51 (46.8) | 136 (64.8) | 0.00 |
| Color and lighting condition | 116 (69.1) | 77 (51.0) | 0.00 | 45 (41.3) | 148 (70.5) | 0.00 |
| Ventilation | 127 (75.6) | 82 (54.3) | 0.00 | 46 (42.2) | 163 (77.6) | 0.00 |
| Aesthetic workplace | 75 (44.6) | 53 (35.1) | 0.08 | 18 (16.5) | 11 (52.4) | 0.00 |
| Washroom facility | 78 (46.4) | 54 (35.8) | 0.05 | 25 (22.9) | 107 (51.0) | 0.00 |
| Noise and sound-free workplace | 106 (63.1) | 77 (51.0) | 0.03 | 50 (45.9) | 133 (63.3) | 0.00 |
| Uninterrupted power supply | 90 (53.6) | 90 (59.6) | 0.28 | 45 (41.3) | 135 (64.3) | 0.00 |
| Uninterrupted water supply | 133 (79.2) | 101 (66.9) | 0.01 | 70 (64.2) | 164 (78.1) | 0.01 |
| Damp-free workplace | 68.5 (115) | 58.3 (88) | 0.06 | 45.9 (50) | 153 (72.9) | 0.00 |
| Respect to opinions | 108 (64.3) | 87 (57.6) | 0.22 | 61 (56.0) | 134 (63.8) | 0.17 |
| Valuation of work | 119 (70.8) | 97 (64.2) | 0.21 | 66 (60.6) | 150 (71.4) | 0.05 |
| Mentoring | 133 (79.2) | 109 (72.2) | 0.15 | 78 (71.6) | 164 (78.1) | 0.20 |
| On-time salary | 154 (91.7) | 147 (97.4) | 0.03 | 100 (91.7) | 201 (95.7) | 0.45 |
| Medicine and MSR supply | 110 (65.5) | 117 (77.5) | 0.02 | 56 (51.4) | 171 (81.4) | 0.00 |

Note:

• MSR: Medical and Surgical Requisites

• Wilcoxon rank-sum test to compare the responses of two groups

• Kruskal-Wallis test to compare the responses of three or more groups

The results in Table 3 showed that the mean weight scores were more than 90 out of 100 for all the indicators, meaning that all the indicators are very vital for delivering quality healthcare. In terms of importance, receiving monthly salary in due time was top-ranked (99.15) to provide quality healthcare, followed by consistent availability of medicines and MSR (98.04), adequate mentoring and support to perform duties (97.57), good relations with co-workers (97.50), and uninterrupted power supply (97.41).

Table 6 showed that women and older workforces tended to assign greater importance to each indicator compared to men and younger workforces, respectively. However, none of these differences are statistically significant. Physicians were less likely than other clinical staff to attribute greater importance to all indicators, with significant differences found only in the following indicators: workplace layout, provider's safety, aesthetic views of the workplace, and noise and sound-free workplace. Pharmacists assigned higher importance than any other

**Table 5. Percentage rating working conditions as good (represented by percent of respondents who rated the indicators as 'agree' and 'strongly agree') for different indicators by gender, age group, years of service in the public hospital, and workforce categories.**

| Indicators | Gender | | | Age | | | Duration of service in the public health facility | | | | Workforce category | | | | | | |
|---|---|---|---|---|---|---|---|---|---|---|---|---|---|---|---|---|---|
| | Male | Female | P-value | <40 | >=40 | P-value | 1–5 | 6–10 | Above 10 | P-value | MO, EMO, RMO | Consultant | Nurse & Midwife | SACMO | Medical technologist | Pharmacist | P-value |
| | n (%) | n (%) | | n (%) | n (%) | | n (%) | n (%) | n (%) | | n (%) | n (%) | n (%) | n (%) | n (%) | n (%) | |
| Work volume | 62 (46.3) | 118 (63.8) | 0.00 | 108 (55.7) | 72 (57.6) | 0.73 | 81 (55.9) | 29 (55.8) | 70 (57.4) | 0.96 | 43 (48.9) | 10 (47.6) | 91 (68.9) | 9 (52.9) | 20 (45.5) | 7 (41.2) | 0.01 |
| Enough staff | 18 (13.4) | 70 (37.8) | 0.00 | 46 (23.7) | 42 (33.6) | 0.05 | 37 (25.5) | 11 (21.2) | 40 (32.8) | 0.22 | 9 (10.2) | 1 (4.8) | 60 (45.5) | 4 (23.5) | 11 (25.0) | 3 (17.7) | 0.00 |
| Job roles and responsibilities | 78 (58.2) | 115 (62.2) | 0.48 | 105 (54.1) | 88 (70.4) | 0.00 | 79 (54.5) | 30 (57.7) | 84 (68.9) | 0.05 | 40 (45.5) | 12 (56.1) | 87 (65.9) | 13 (76.5) | 30 (68.2) | 11 (64.7) | 0.02 |
| Required training received | 61 (45.5) | 64 (34.6) | 0.05 | 68 (35.1) | 57 (45.6) | 0.06 | 48 (33.1) | 24 (46.2) | 53 (43.4) | 0.12 | 33 (37.5) | 16 (76.2) | 44 (33.3) | 10 (58.8) | 19 (43.2) | 3 (17.7) | 0.00 |
| Compatibility of rank | 61 (45.5) | 116 (62.7) | 0.00 | 127 (65.5) | 50 (40.0) | 0.00 | 102 (70.3) | 29 (55.8) | 46 (37.7) | 0.00 | 60 (68.2) | 13 (61.9) | 84 (63.6) | 5 (29.4) | 11 (25.0) | 4 (23.5) | 0.00 |
| Workplace layout | 41 (30.6) | 73 (39.5) | 0.10 | 69 (35.6) | 45 (36.0) | 0.94 | 47 (32.4) | 19 (36.6) | 48 (39.3) | 0.50 | 32 (36.4) | 5 (23.8) | 44 (33.3) | 9 (52.9) | 16 (36.4) | 8 (47.1) | 0.44 |
| Functional medical equipment | 47 (35.1) | 102 (55.1) | 0.00 | 85 (43.8) | 64 (51.2) | 0.20 | 63 (43.4) | 25 (48.1) | 61 (50.0) | 0.55 | 24 (27.3) | 4 (19.1) | 81 (61.4) | 8 (47.1) | 24 (54.6) | 8 (47.1) | 0.00 |
| Provider's safety | 37 (27.6) | 65 (35.1) | 0.16 | 55 (28.4) | 47 (37.6) | 0.08 | 37 (25.5) | 16 (30.8) | 49 (40.2) | 0.04 | 11 (12.5) | 1 (4.8) | 45 (34.1) | 8 (47.1) | 27 (61.4) | 10 (58.8) | 0.00 |
| Stress-free workplace | 33 (24.6) | 66 (35.7) | 0.04 | 58 (29.9) | 41 (32.8) | 0.58 | 41 (28.3) | 17 (32.7) | 41 (33.6) | 0.62 | 16 (18.2) | 3 (14.3) | 50 (37.9) | 9 (52.9) | 16 (36.4) | 5 (29.4) | 0.01 |
| Burnout | 41 (30.6) | 79 (42.7) | 0.03 | 83 (42.8) | 37 (29.6) | 0.02 | 67 (46.2) | 19 (36.5) | 34 (27.9) | 0.01 | 38 (43.2) | 6 (28.6) | 61 (46.2) | 1 (5.9) | 8 (18.2) | 6 (35.3) | 0.00 |
| Recognition of work | 53 (39.6) | 106 (57.3) | 0.00 | 90 (46.4) | 69 (55.2) | 0.13 | 70 (48.3) | 23 (44.2) | 66 (54.1) | 0.43 | 28 (31.8) | 6 (28.6) | 77 (58.3) | 8 (47.1) | 28 (63.6) | 12 (70.6) | 0.00 |
| Relationship with co-workers | 127 (94.8) | 180 (97.3) | 0.24 | 187 (96.4) | 120 (96.0) | 0.86 | 139 (95.9) | 51 (98.1) | 117 (95.9) | 0.75 | 85 (96.6) | 19 (90.5) | 128 (97.0) | 15 (88.2) | 43 (97.7) | 17 (100.0) | 0.30 |
| Clean workplace | 78 (58.2) | 109 (58.9) | 0.90 | 108 (55.7) | 79 (63.2) | 0.18 | 76 (52.4) | 34 (65.4) | 77 (63.1) | 0.12 | 40 (45.5) | 11 (52.4) | 82 (62.1) | 9 (52.9) | 34 (77.3) | 11 (64.7) | 0.02 |
| Color and lighting condition | 78 (58.2) | 115 (62.2) | 0.48 | 111 (57.2) | 82 (65.6) | 0.14 | 81 (55.9) | 30 (56.7) | 82 (67.2) | 0.15 | 40 (45.4) | 5 (23.8) | 86 (65.2) | 11 (64.7) | 39 (88.7) | 12 (70.6) | 0.00 |
| Ventilation | 78 (58.2) | 131 (70.8) | 0.02 | 123 (63.4) | 86 (68.8) | 0.32 | 90 (62.1) | 34 (65.4) | 85 (69.7) | 0.42 | 39 (44.3) | 7 (33.3) | 96 (72.7) | 15 (88.2) | 39 (88.6) | 13 (76.5) | 0.00 |
| Aesthetic workplace | 48 (35.8) | 80 (43.2) | 0.18 | 64 (33.0) | 51 (51.2) | 0.00 | 43 (29.7) | 21 (40.4) | 64 (52.5) | 0.00 | 17 (19.3) | 1 (4.8) | 59 (44.7) | 13 (76.5) | 28 (66.6) | 10 (58.8) | 0.00 |
| Washroom facility | 57 (42.5) | 75 (40.5) | 0.72 | 71 (36.6) | 61 (48.8) | 0.03 | 47 (32.4) | 23 (44.2) | 62 (50.8) | 0.01 | 23 (26.1) | 2 (9.52) | 59 (44.7) | 8 (47.1) | 28 (63.6) | 12 (70.6) | 0.00 |
| Noise and sound-free workplace | 77 (57.5) | 106 (57.3) | 0.98 | 107 (55.2) | 76 (60.8) | 0.32 | 73 (50.3) | 31 (59.6) | 79 (64.8) | 0.06 | 41 (46.6) | 9 (42.9) | 76 (57.6) | 12 (70.6) | 34 (77.3) | 11 (64.7) | 0.01 |
| Uninterrupted power supply | 80 (59.7) | 100 (54.1) | 0.32 | 100 (51.6) | 80 (64.0) | 0.03 | 70 (48.3) | 29 (55.8) | 81 (66.4) | 0.01 | 41 (46.6) | 4 (19.1) | 75 (56.8) | 11 (64.7) | 37 (84.1) | 12 (70.6) | 0.00 |
| Uninterrupted water supply | 105 (78.4) | 129 (69.7) | 0.09 | 139 (71.7) | 95 (76.0) | 0.39 | 100 (69.0) | 40 (76.9) | 94 (77.1) | 0.27 | 61 (69.3) | 9 (42.9) | 95 (72.0) | 14 (82.4) | 41 (93.2) | 14 (82.4) | 0.00 |
| Damp-free workplace | 81 (60.5) | 122 (66.0) | 0.32 | 123 (63.4) | 80 (64.0) | 0.91 | 88 (60.7) | 33 (63.5) | 82 (67.2) | 0.54 | 43 (48.9) | 7 (33.3) | 93 (70.5) | 12 (70.6) | 36 (81.8) | 12 (70.6) | 0.00 |
| Respect to opinions | 82 (61.2) | 113 (61.1) | 0.98 | 109 (56.2) | 86 (68.8) | 0.02 | 83 (57.2) | 32 (61.5) | 80 (65.6) | 0.38 | 48 (54.6) | 13 (61.9) | 85 (64.4) | 9 (52.9) | 29 (65.9) | 11 (64.7) | 0.67 |
| Valuation of work | 88 (65.7) | 128 (69.2) | 0.51 | 128 (66.0) | 88 (70.4) | 0.41 | 95 (65.5) | 38 (73.1) | 83 (68.0) | 0.60 | 55 (62.5) | 11 (52.4) | 93 (70.5) | 9 (52.9) | 36 (81.8) | 12 (70.6) | 0.09 |
| Mentoring | 103 (76.9) | 139 (75.1) | 0.72 | 139 (71.7) | 103 (82.4) | 0.03 | 101 (69.7) | 40 (76.9) | 101 (82.8) | 0.04 | 63 (71.6) | 15 (71.4) | 100 (75.8) | 14 (82.4) | 38 (86.4) | 12 (70.6) | 0.49 |
| On time salary | 123 (91.8) | 178 (96.2) | 0.09 | 179 (92.3) | 122 (97.6) | 0.04 | 130 (89.7) | 51 (98.1) | 120 (98.4) | 0.00 | 81 (92.1) | 19 (90.5) | 127 (96.2) | 16 (94.1) | 42 (95.5) | 16 (94.1) | 0.79 |
| Medicine and MSR supply | 85 (63.4) | 142 (76.7) | 0.01 | 127 (65.5) | 100 (80.0) | 0.01 | 89 (61.4) | 38 (73.1) | 100 (82.0) | 0.00 | 45 (51.1) | 11 (52.4) | 108 (81.8) | 15 (88.2) | 34 (77.3) | 14 (82.4) | 0.00 |

Note:

• MO: Medical Officer

• EMO: Emergency Medical Officer

• RMO: Residential Medical Officer

• SACMO: Sub-Assistant Community Medical Officer

• MSR: Medical and Surgical Requisites

• Wilcoxon rank-sum test to compare the responses of two groups

• Kruskal-Wallis test to compare the responses of three or more groups

**Table 6. Mean weight score of different indicators of working conditions regarding respondents' background characteristics.**

| Indicators | Gender | | Age | | Workforce type | | Workforce Category | | | | | | Duration of service in the public hospital | | |
|---|---|---|---|---|---|---|---|---|---|---|---|---|---|---|---|
| | Male | Female | Less than 40 | 40 and above | Physician | Other clinical staff | MO, EMO, and RMO | Consultant | Nurse and Midwife | SACMO | Medical Technologist | Pharmacist | 1–5 Years | 6–10 Years | Above 10 Years |
| Work volume | 93.6 | 96.5 | 95.1 | 95.5 | 91.9 | 97.0 | 91.4 | 94.2 | 97.4 | 97.0 | 95.9 | 97.0 | 95.2 | 94.6 | 95.7 |
| Enough staff | 95.1 | 96.6 | 95.8 | 96.2 | 92.0 | 98.0 | 92.1 | 91.9 | 98.3 | 99.4 | 96.0 | 99.4 | 96.2 | 93.7 | 96.6 |
| Job roles and responsibilities | 94.8 | 96.8 | 95.5 | 96.7 | 91.8 | 98.1 | 91.5 | 92.8 | 98.2 | 94.4 | 99.0 | 98.8 | 95.1 | 95.0 | 97.3 |
| Required training received | 94.6 | 96.7 | 94.8 | 97.4 | 92.6 | 97.5 | 92.6 | 92.3 | 97.3 | 94.7 | 98.4 | 99.4 | 95.0 | 94.0 | 97.5 |
| Compatibility of rank | 95.8 | 97.3 | 96.2 | 97.4 | 92.8 | 98.7 | 92.6 | 93.3 | 98.6 | 97.9 | 98.8 | 100.0 | 96.6 | 94.1 | 97.9 |
| Workplace layout | 91.5 | 95.3 | 92.7 | 95.2 | 88.4 | 96.4 | 89.0 | 86.1 | 96.2 | 95.5 | 96.7 | 97.9 | 92.8 | 91.9 | 95.5 |
| Functional medical equipment | 95.5 | 97.9 | 96.3 | 97.8 | 92.6 | 99.1 | 92.2 | 94.2 | 99.2 | 99.4 | 98.6 | 100.0 | 96.3 | 96.0 | 98.0 |
| Provider's safety | 94.4 | 97.3 | 95.9 | 96.4 | 90.6 | 99.0 | 90.4 | 91.4 | 98.8 | 99.7 | 98.8 | 100.0 | 96.3 | 93.4 | 97.0 |
| Stress-free workplace | 94.4 | 96.7 | 95.6 | 96.0 | 91.2 | 98.1 | 92.1 | 87.6 | 98.5 | 97.6 | 96.8 | 98.2 | 96.2 | 94.5 | 95.7 |
| Burnout | 95.0 | 97.2 | 95.5 | 97.5 | 91.7 | 98.7 | 90.9 | 95.2 | 98.4 | 98.2 | 99.5 | 99.4 | 96.1 | 93.1 | 97.8 |
| Recognition of work | 94.7 | 96.8 | 95.3 | 96.9 | 91.5 | 98.2 | 91.7 | 90.4 | 98.3 | 96.4 | 98.0 | 100.0 | 95.9 | 92.9 | 97.2 |
| Relationship with co-workers | 97.1 | 97.7 | 97.3 | 97.7 | 94.4 | 99.0 | 95.2 | 91.4 | 98.8 | 98.5 | 99.6 | 100.0 | 97.1 | 97.6 | 97.9 |
| Clean workplace | 96.2 | 97.1 | 96.3 | 97.4 | 92.2 | 99.1 | 92.7 | 90.4 | 98.8 | 100.0 | 99.2 | 99.7 | 96.0 | 96.9 | 97.6 |
| Color and lighting condition | 93.1 | 94.8 | 92.9 | 95.8 | 89.0 | 96.7 | 89.5 | 87.1 | 95.9 | 96.1 | 98.0 | 100.0 | 92.5 | 94.3 | 95.8 |
| Ventilation | 94.9 | 94.7 | 95.4 | 96.9 | 91.1 | 98.5 | 91.8 | 88.3 | 98.2 | 96.7 | 99.5 | 97.0 | 95.8 | 94.2 | 97.0 |
| Aesthetic workplace | 88.4 | 93.3 | 90.3 | 92.7 | 83.1 | 95.5 | 83.9 | 79.7 | 94.9 | 96.1 | 96.5 | 100.0 | 90.8 | 88.3 | 93.0 |
| Washroom facility | 95.2 | 97.0 | 95.8 | 96.9 | 91.4 | 98.7 | 91.9 | 89.0 | 98.4 | 100.0 | 98.7 | 100.0 | 96.1 | 95.1 | 96.8 |
| Noise and sound-free workplace | 94.8 | 96.3 | 95.3 | 96.2 | 90.0 | 98.6 | 90.4 | 88.5 | 98.0 | 100.0 | 99.5 | 100.0 | 96.0 | 92.7 | 96.6 |
| Uninterrupted power supply | 97.0 | 97.6 | 97.2 | 97.7 | 93.3 | 99.5 | 94.0 | 90.4 | 99.3 | 100.0 | 99.5 | 100.0 | 97.3 | 97.1 | 97.6 |
| Uninterrupted water supply | 97.2 | 97.3 | 97.1 | 97.5 | 92.6 | 99.7 | 93.1 | 90.4 | 99.5 | 100.0 | 100.0 | 100.0 | 97.5 | 95.9 | 97.4 |
| Damp-free workplace | 95.2 | 97.3 | 96.1 | 96.9 | 91.4 | 99.0 | 92.0 | 89.0 | 99.1 | 98.8 | 98.4 | 99.4 | 96.1 | 96.3 | 96.9 |
| Respect to opinions | 95.1 | 97.3 | 96.0 | 97.0 | 92.1 | 98.6 | 92.3 | 91.4 | 98.8 | 99.1 | 97.7 | 100.0 | 96.2 | 95.2 | 97.1 |
| Valuation of work | 96.6 | 97.4 | 96.7 | 97.7 | 93.2 | 991 | 93.5 | 92.3 | 99.0 | 97.9 | 99.3 | 100.0 | 97.2 | 95.5 | 97.6 |
| Mentoring | 96.7 | 98.1 | 97.6 | 97.4 | 94.0 | 99.3 | 94.6 | 91.6 | 99.4 | 99.4 | 99.0 | 100.0 | 97.7 | 96.9 | 97.6 |
| On-time salary | 99.2 | 99.0 | 99.1 | 99.1 | 07.6 | 99.9 | 97.8 | 96.6 | 99.9 | 100.0 | 100.0 | 100.0 | 98.9 | 99.4 | 99.2 |
| Medicine and MSR supply | 97.3 | 98.5 | 97.8 | 98.3 | 95.0 | 99.5 | 94.9 | 95.4 | 99.5 | 99.4 | 99.5 | 100.0 | 98.0 | 97.1 | 98.4 |

Note:

• MO: Medical Officer

• EMO: Emergency Medical Officer

• RMO: Residential Medical Officer

• SACMO: Sub-Assistant Community Medical Officer

• MSR: Medical and Surgical Requisites

health workforce category to all indicators except work volume, stress-free workplace, clean workplace, and ventilation. Nurses and midwives gave greater importance than other workforce categories to work volume and a stress-free workplace, while medical technologists prioritized workplace ventilation and SACMOs prioritized a clean workplace. Nevertheless, none of these differences were statistically significant.

Table 7 presented the psychometric analysis results for the indicators of working conditions. The findings revealed that most indicators had substantial factor loadings (0.50 or higher), indicating a strong correlation between survey responses and the latent construct of working conditions. This provided strong support for the validity of using these indicators to assess working conditions. Additionally, the uniqueness values, which indicate the explanatory power of the factor, were moderately high for most indicators. For instance, the uniqueness value for the statement "color and lighting condition is compatible to work" was 0.47, indicating that the factor explains 53% of the variance in responses related to working conditions. Consequently, we concluded that the majority of the indicators can be considered as components of a single construct, denoted as the working conditions of clinical health workforces. Moreover, the values of Cronbach's alpha, which assess the unidimensionality of the indicators, were nearly 0.90 for all indicators, confirming that they measured a unidimensional construct effectively.

**Table 7. Confirmatory factor analysis (CFA) and Cronbach alpha coefficients estimated from the indicators of the working conditions.**

| Indicators | Factor analysis | | Cronbach alpha coefficient | |
|---|---|---|---|---|
| | Factor loadings | Uniqueness | Interitem correlation | Cronbach alpha |
| Work volume | 0.31 | 0.90 | 0.26 | 0.90 |
| Enough staff | 0.44 | 0.80 | 0.25 | 0.90 |
| Job roles and responsibilities | 0.49 | 0. 75 | 0.25 | 0.90 |
| Required training received | 0.25 | 0.93 | 0.26 | 0.90 |
| Compatibility of rank | 0.05 | 0.99 | 0.27 | 0.90 |
| Workplace layout | 0.49 | 0.76 | 0.25 | 0.90 |
| Functional medical equipment | 0.60 | 0.64 | 0.25 | 0.90 |
| Provider's safety | 0.52 | 0.72 | 0.25 | 0.90 |
| Stress-free workplace | 0.50 | 0.74 | 0.25 | 0.90 |
| Burnout | 0.29 | 0.91 | 0.25 | 0.90 |
| Recognition of work | 0.51 | 0.73 | 0.26 | 0.90 |
| Relationship with co-workers | 0.40 | 0.84 | 0.25 | 0.90 |
| Clean workplace | 0.70 | 0.51 | 0.25 | 0.90 |
| Color and lighting condition | 0.72 | 0.47 | 0.25 | 0.90 |
| Ventilation | 0.70 | 0.50 | 0.25 | 0.89 |
| Aesthetic workplace | 0.66 | 0.57 | 0.25 | 0.89 |
| Washroom facility | 0.69 | 0.51 | 0.25 | 0.89 |
| Noise and sound-free workplace | 0.58 | 0.66 | 0.25 | 0.89 |
| Uninterrupted power supply | 0.53 | 0.71 | 0.25 | 0.89 |
| Uninterrupted water supply | 0.64 | 0.58 | 0.25 | 0.89 |
| Damp-free workplace | 0.65 | 0.57 | 0.25 | 0.90 |
| Respect to opinions | 0.57 | 0.67 | 0.25 | 0.89 |
| Valuation of work | 0.57 | 0.67 | 0.25 | 0.89 |
| Mentoring | 0.58 | 0.66 | 0.25 | 0.89 |
| On-time salary | 0.27 | 0.92 | 0.26 | 0.90 |
| Medicine and MSR supply | 0.54 | 0.69 | 0.25 | 0.90 |

## 4. Discussions

The findings suggested that the overall working conditions of clinical health workforces were quite poor, particularly in secondary-level facilities. Similar findings were observed in a study in South Africa where it showed the status of working conditions in the public hospitals was poor, which was characterized by job dissatisfaction, poor infrastructure, stress, and burnout [29]. Our findings indicated that poor working conditions were influenced by several factors, including but not limited to the absence of safety measures to protect healthcare providers against violence, unsuitable workplace layout design, inadequate healthcare workforce, stress, insufficient training, and the absence of promotion opportunities. However, some high-performing indicators, such as receiving salary in due time, relationships with co-workers, and proper mentoring, were also identified in our study.

One of the significant findings of this study was that the likelihood of reporting good working conditions was higher in primary than secondary level facilities in 11 out of 26 indicators. Additionally, physicians tended to report poorer working conditions compared to other clinical staff. Moreover, most other clinical staff members claimed that their job rank do not correspond to their length of service. These findings were supported by a number of previous studies in different settings. For instance, a study in Tanzania found that healthcare workers in primary healthcare facilities experienced better working conditions than those in district hospitals [30]. Similarly, two different studies in Ghana, and Ethiopia explored that physician reported significantly lower job satisfaction compared to other healthcare professionals [31, 32]. Moreover, nurses and clinical officers retorted lack of promotion opportunities and recognition for their work in a study in Ethiopia [32]. The implications of these findings are significant and suggest poor working conditions among physicians may lead to increase job dissatisfaction, stress, and burnout, which in turn could negatively impact patient care. Improving working conditions for physicians and other clinical staff by creating merit-based promotion opportunities could improve staff retention and motivation. In addition, investment in improving working conditions at the secondary level facilities may be needed to meet required standards, such as addressing workforce shortages and improving workplace infrastructure and layout.

The study also revealed that men, younger workforce, respondents with MBBS degrees or postgraduate degrees, and workforce with shorter length of service report poor working conditions compared to their counterparts. The results are consistent with previous studies that found male health workers in Nigeria were more likely to experience job dissatisfaction and poorer working conditions than female workforce [33], and that younger health workers in Iran were more likely to report poor job satisfaction and poor working conditions compared to their counterparts [34]. Men may have higher expectations of working conditions than women. However, the study did not find any evidence to suggest that men and women have different expectations of working conditions. The study also highlighted that women did not receive the required training, which could be due to a lack of training opportunities available for women or a result of gender bias in the provision of training. We believe that it is important to further investigate these issues to ensure that all clinical staff, regardless of gender, have equal access to training opportunities to improve the quality of care they provide.

Lastly, findings of assessing importance of different components of working conditions in delivering quality healthcare revealed that timely payment of salaries is the most critical factor, followed by proper recognition of work, enough staff, and adequate functional medical equipment. Previous studies supported the importance of salary in motivating healthcare workers, as well as recognition, adequate staffing, and functional equipment. Three studies conducted separately in Ethiopia, Nigeria, and Ghana found that healthcare workers identify the timely

payment, recognition and appreciation of work, and adequate functional medical equipment, respectively, as the most critical factors for improving their job satisfaction and motivation to provide quality care [32, 35, 36]. Therefore, neglecting these critical factors may lead to demotivation and burnout among healthcare workers, ultimately resulting in decreased productivity, absenteeism, and compromised quality of care. Addressing these factors is crucial to ensure a supportive and productive working environment for healthcare workers and the delivery of quality care.

The findings of this study have important implications for policymakers and healthcare managers in Bangladesh. There is a need for immediate interventions to improve the working conditions of healthcare workers, particularly in secondary level facilities. The study suggests that efforts should be made to provide better working conditions to physicians and male healthcare workers. In addition, the study highlights the importance of giving salary in due time, consistent supply of medicine and MSR, and adequate mentoring to provide quality care.

## 5. Conclusion

It is evident that the working conditions of the clinical health workforce in public health facilities in Bangladesh are poor. Overall, improving working conditions in healthcare facilities requires a comprehensive approach that addresses the various factors that affect the healthcare workforce. Although the findings may not be generalized due to the absence of nationally representative sampling, these have important policy implications. The study suggests that policymakers should take necessary steps to improve working conditions in both types of facilities specially, in secondary level facilities. However, priority should be given to recruiting more health workforce, adopting evidence-based safety measures against violence in the workplace, designing hospital layout so that it does not hamper work efficiency, and eliminating deprivation in promotion and higher education. Further research with a larger size incorporating all the administrative divisions of the country is crucial for assessing working conditions more comprehensively to generalize the results. However, by implementing these strategies, governments and public sectors in developing countries specially, Bangladesh can improve the working conditions of health workers, which can lead to improved healthcare outcomes and better patient experiences.

## 6. Limitations

Overall, while the study provides valuable insights into the working conditions of health workers in Bangladesh, its limitations should be taken into account when interpreting the results and drawing conclusions. The following are some potential limitations of the study.

Small sample size: Due to resource constraints, the study was limited to a small number of samples and covered only four out of eight administrative divisions of the country.

Use of CFA: Given our small sample size, conducting confirmatory factor analysis (CFA) may lead to unstable estimates, potentially influencing the reliability of our construct validity assessment.

Self-reported data: The study relied on self-reported data from participants, which may be subject to bias. For example, participants may be more likely to report positive working conditions to avoid repercussions or negative feedback. Similarly, participants may not accurately remember or understand their working conditions, which could affect the validity of the results.

The limited scope of indicators: The study measured working conditions across 26 indicators, which may not capture the full range of factors that contribute to quality healthcare. For

example, the study did not measure the impact of working conditions on patient outcomes, which could be an important consideration in evaluating the quality of care.

Limited generalizability: The study focused on healthcare workers in public facilities in Bangladesh, which may not be representative of healthcare workers in other settings or countries. Therefore, the findings may not be generalizable beyond this specific context.

## Supporting information

**S1 Data.**
(XLSX)

## Acknowledgments

The authors expressed their gratitude to the respondents who participated in the survey by providing their valuable time.

## Author Contributions

**Conceptualization:** Syed Abdul Hamid, Md. Ragaul Azim.

**Data curation:** Md. Mahfujur Rahman, Md. Sirajul Islam.

**Formal analysis:** Md. Mahfujur Rahman, Md. Sirajul Islam.

**Funding acquisition:** Syed Abdul Hamid, Md. Ragaul Azim.

**Investigation:** Md. Mahfujur Rahman, Md. Sirajul Islam.

**Methodology:** Syed Abdul Hamid, Md. Ragaul Azim.

**Project administration:** Md. Mahfujur Rahman, Md. Sirajul Islam.

**Supervision:** Syed Abdul Hamid, Md. Ragaul Azim.

**Writing – original draft:** Syed Abdul Hamid, Md. Ragaul Azim, Md. Mahfujur Rahman, Md. Sirajul Islam.

**Writing – review & editing:** Syed Abdul Hamid, Md. Ragaul Azim.

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
