## [Decision Letter · Decision Letter 0]

6 Mar 2023

PONE-D-22-23402Working conditions of the clinical health workforce in the public health facilities in BangladeshPLOS ONE

Dear Dr. Azim,

Thank you for submitting your manuscript to PLOS ONE. After careful consideration, we feel that it has merit but does not fully meet PLOS ONE’s publication criteria as it currently stands. Therefore, we invite you to submit a revised version of the manuscript that addresses the points raised during the review process.

We look forward to receiving your revised manuscript.

Kind regards,

Humayun Kabir, MSc in Epidemiology

Academic Editor

PLOS ONE

Journal Requirements:

Reviewers' comments:

Reviewer's Responses to Questions

**Comments to the Author**

1. Is the manuscript technically sound, and do the data support the conclusions?

Reviewer #1: Yes

Reviewer #2: No

Reviewer #3: Yes

Reviewer #4: No

Reviewer #5: No

2. Has the statistical analysis been performed appropriately and rigorously? 

Reviewer #1: Yes

Reviewer #2: No

Reviewer #3: Yes

Reviewer #4: Yes

Reviewer #5: No

3. Have the authors made all data underlying the findings in their manuscript fully available?

Reviewer #1: Yes

Reviewer #2: Yes

Reviewer #3: No

Reviewer #4: Yes

Reviewer #5: No

4. Is the manuscript presented in an intelligible fashion and written in standard English?

Reviewer #1: Yes

Reviewer #2: No

Reviewer #3: Yes

Reviewer #4: No

Reviewer #5: No

5. Review Comments to the Author

Reviewer #1: There are a few comments I have made that if addressed could improve the manuscript which explores the working conditions of clinical of workers in public health facilities in Bangladesh. The authors have undertaken a survey based upon random selection from four secondary level and age primary level public health care facilities and administered a pretested survey. They describe what they consider poor working conditions using 26 indicators. They also undertook to see which working conditions considered by the health workforce as essential for the delivery of quality safe care.

Overall, the English is good, although there is some overuse of superlatives and the occasional quaint expression. This is not affecting the meaning but could readily be addressed by some editing. e.g. “Comprehensively measured scientific evidence on the level of working conditions of the health workforce is highly imperative for appealing effective policy interventions”

There is a quite an elaborate sampling strategy described in the methods but very little is made of this in the results. I wondered whether how the sampling strategy earns its place in the paper when so little is made of it in terms of the analysis and reporting. Was there a theory to justify the sampling strategy? Were there differences between the four districts and eight upazilas?

Who is in the study (i.e. the participants) is usually provided in the results not the methods. The following description of participants should be moved the results, “The study includes 45 physicians and 123 other clinical staff from the UHCs, and 64 physicians and 87 other clinical staff from the DHs using a random sampling technique. Thus, we interview a total of 319 clinical staff (physicians: 109, other clinical staff: 210).

The paper describes how they dealt with neutral in a Likert-type response. “Following the literature, the study considers the neutral option along with disagreeing and strongly disagree options for explaining poor working conditions because people often select the neutral option to respond indifferently or not positively (15,16). Refs 15 and 16 are papers written by the first author and the first refence only states much the same thing as is stated in the paper under review, without any referencing to support their statement “We consider the neutral option along with disagreeing and strongly disagree options for defining negative perception. The reason is that usually, an individual in Bangladesh chooses the neutral option if she likes to respond indifferently or not positively.” The second paper does not even refer to neutral responses. The second reference (that is number 16) should certainly be removed. Reference #15 provides no more no support for the assertion other than that they have used this approach before. It is important that authors describe how they have dealt with the data but they should not dress it up as it being based upon evidence which if it does exist is not provided.

15. Hamid SA, Begum A, Azim MR, Islam MS. Doctor-patient relationship: Evidence from Bangladesh. Health Science Reports. 2021 Dec 1;4(4).

16. Hamid SA, Begum A. Responsiveness of the urban primary health care delivery system in Bangladesh: A comparative analysis. International Journal of Health Planning and Management. 2019 Jan 1;34(1):251–62.

The very high mean weights for the importance of the work characteristics shows that almost all participants ranked the items as very important or important. Of course, there is a lot wider distribution of responses to the ranking of their current workplace environment.

The Results should open with the overall description of the respondents (moved from the methods above)

It is not wrong but clunky saying “significant (p-value ≤.05)” each time – conventionally, these things are reported just as (p≤0.05)

Most people aged 40 years or above would not be happy to be called elderly. Suggest change “elderly” to “older” throughout.

The authors report that they did consult the literature in developing their list of various working conditions. Almost all of the factors were considered important by the respondents. Being paid on time and the relationship with co-workers were reported as the most important determinants for a good working environment. However, it may be considered is a weakness in a study design to just provide a list without requiring any process of deciding the relative importance of them i.e. ranking their importance, rather than that are very important.

The authors state that the magnitude of working conditions is comparatively poor for men younger workforce and some others. However it is not clear whether the conditions are different for them or whether the men are simply are more critical of the conditions. Perhaps men for example have higher expectations then women? it is also interesting that women felt they were not being able to access the training that they need and again it should be discussed in whether there is a gender bias in what training is available for women or whether women just feel they need additional training in order to feel competent in delivering care. These possibilities should be discussed.

The authors have made an original contribution and one important for maintaining a clinical health workforce in Bangladesh. The manuscripts seems technically sound

Reviewer #2: PLOS One

Title: Working conditions of the clinical health workforce in the public health facilities in Bangladesh.

Comments to the authors:

General comments:

- This study lacking is novelty, specificity in objectives and unplanned statistical analysis with improper interpretation.

- The conclusion based on imprecise calculation of a subjective variable (weighted mean score??).

Specific comments:

Abstract section:

- There is no clear objective of this study, merely to trigger policy discussion? Improving efficiency and effectiveness of the healthcare service delivery system is beyond the scope of this paper.

- The score (weighted mean score) of 3.4 (how? and why?) was considered to represent poor working conditions?

- The average (mean?) weight score (out of 100 points) indicating that all parameters included were pre-requisites for delivering effective and efficient services??? This is very confusing and contradictory.

- The conclusion section did not reflect the actual work done in this research.

Introduction section:

- This section is lacking the literature related to work place and working condition in Bangladesh and other developing countries and the reflections on quality of services and staff.

- Working condition and how significant in terms of efficiency, effectiveness and satisfaction of the health care team.

- What kind of strategy adopted and lunched by the government and public sectors to improve the work conditions in Bangladesh and other developing countries with similar circumstances?

- “Although a growing body of the literature focuses on defining the components of working conditions of the health workforce, there is no scientific research globally on the status of working conditions measured comprehensively” this is not correct and needs references

- There is no concrete rationale for the study.

- The objective of this study was simply to assess the working conditions at primary and secondary care setting and to estimate the relative importance of each dimension form the providers’ perspectives.

- Line 88 to line 90: remove

Materials and methods (replaced by subjects and methods):

- Replace data with sampling methods

- What was your study design?

- Setting should be mentioned (what is upazilas? the second lower tire of administrative structure in Bangladesh) should be defined, and how many each can serve.

- Your sampling methods should be proposed as a Multi-stage sampling methods with each stage should be provided in details: first how many districts were included in the primary selection (4 districts were randomly selected out of ----) for the secondary care centers and 8 upazilas (randomly selected from ------) form the greater administrative divisions to represent the primary care facilities.

- The second stage is the selection of interviewees (systematic sampling method should be followed, database or list of all clinical staff in the selected centers should be available for that purpose)

- Sample size should be determined before selection of participants (not found), sample size should also consider the segmentation and variations in respondents (physicians, nurses, etc.,)

Data collection:

- Based on literature review you had consulted, what were the pros and cons for each definition of working conditions, how was each instrument reliable and valid?

- Line 117: not clear

- How did you invite participants? was it a personal interviews and by whom? And in what language the data collection form was administered?

- How many responded to your invitation? how many refused/declined? were they different form the respondents?

- Table 1:

Item 1: subjective

Item 2: enough staff?? How many is the ideal?

Item 3: roles vs. responsibility (double barrel question)

Item 5: define efficiency

I suggest to categorize these items into fewer dimensions: work performance indicators (volume, training, equipment, and staffing), psychological (rewarding, protection, recognition, burn out and etc., ), and physical working conditions (layout, color and lighting, water, sanitation etc.,)

How the questionnaire customized? “In addition, the study measures the same indicators on a scale of 100 to assess weight based on their relative importance to provide quality healthcare”. Why and how?

Data analysis:

- Replace estimates with calculates (considered the tense)

- Express your numerical variables using median, interquartile range and mean.

- This section is completely not clear “The study calculates a weighted mean score adjusting the weight given by the respondents to each indicator with their reported score to calculate the overall status of the working conditions.

- There is should a valid method to determine this cut-off (using percentiles), We define the working conditions as good when the weighted mean score is 4 or above, and poor when it is below 4. Finally, we use the chi-square test to assess whether the responses of the different categories of respondents are significantly different or not. Chi-square for which variables?

- From line 155 to line 165: this section is totally incorrect and should be removed from the manuscript.

- This proposed questionnaire should be piloted on a sample beyond the selected candidates to measure reliability if any

- Construct validity should be verified using other method not the confirmatory factor analysis?????

Results:

- Table 2: should display both socio-demographics vs. the responses for each dimensions (and the relevant items).

- Table 3: should display the responses in relation to the type of services (primary vs. secondary)

- Non parametric tests of significance should be used to compare scores in any table displayed.

- This section is so redundant as many parts of the tables mentioned in text of this section.

- Table 5 and 6: very confusing.

- The manuscript is severely disorganized: The study calculates the mean weight score of each indicator to assess their relative importance to provide quality healthcare by the health workforces in the facility. We ask the respondents to put weight on a scale of 100 on each of the indicators of working conditions considering their relative importance to deliver quality healthcare (this section belongs to the methods section)????

Discussion section:

- Start your discussion with providing the answer to your research question!!!!

- The authors mentioned many segments form the results section with repetitions

- Very lengthy discussion with redundancy

Conclusion section:

- There should be a separate section for study limitations which are many

- “The findings of the study are not directly comparable, as there is a paucity of evidence assessing the status of working conditions compared with different types of healthcare facilities and providers in both national and global contexts” this is totally incorrect.

- This section is also confusing and did not reflect the actual work done.

References:

- References to different data collection instruments used in the literature were not mentioned

Reviewer #3: I congratulate the authors for preparing this manuscript on such an important topic that can potentially guide future public health approaches in Bangladesh. The authors have carried out rigorous data analysis. Particularly for Likert-type, ordinal-scale responses, non-parametric statistics (such as chi-square) are typically considered the most appropriate option for analysis.

One minor comment: the authors need to provide evidence to support the statement on p.4, lines 83–84. To my knowledge, there exists comprehensive measures working conditions in the hea;th sector, e,g,, Maassen et al (2021) DOI: 10.1371/journal.pone.0247530; ILO (2017) ISBN 978-92-2-130533-0; Kristensen (2010) DOI: 10.1177/1403494809354437, etc.

Particularly, Kristensen's COPSOQ tool has been adopted and implemented in many countries.

Reviewer #4: The authors have attempted to provide insight into the nature of burden of workload healthcare workers have in that region. This is a diagnostic survey and is considered very appropriate. However, the rigor it require is lacking. My observations are:

1. The title: This appear incomplete considering that the study focused on two major premise (lines 84-86) "assessing the level of working conditions of the health workforce...study also assesses the relative importance of the different factors of working conditions." What factors are these? How these are related should be the burden of this study yet the title has not adequately captured it.

2. The Abstract: The abstract needs to be well articulated for such an important study. The context healthcare in line 36 is missing. The objective of measuring level of working condition appear incomplete.

The description of the data collection instrument is inadequate, line 42: "...administers a customized…", what is customized about the instrument?

The result stated in the abstract reflecting the overall weighted mean score of the working conditions is on a scale of 5 which reflects a poor working condition does not demonstrate such since nothing about the instrument measurement scale is mentioned in the methodology section. What is the scale of measures for variables in the study?

3. Introduction: At a certain stage in the narrative in the introduction should be a presentation of the theoretical and conceptual framework underpinning of what constitute the workload that is to be measured and the antecedent factors relative the the outcomes and consequences in a theoretical clarification. This provides the scientific foundation of the argument in the study.

4. The methodology has some merits but lack coherence. Table 1, shows 26 elements of the working condition measured on a 5-response Likert scale. My concerns are why "neutral", how is neutral scored in relation to others to create a rating scale to measure each participant in the study? What is the final rating scale for this variable? From my judgment, the reference scale for this variable should be a maximum of 104 points representing heavy workload. The cut-off defining threshold to what constitute unacceptable level should be defined and justified.

5. Results: The data analysis require adequate organization and analysis to provide clear answers for the research questions, validity for the findings and conclusion. The presentation of frequency distribution of all items of the instrument without synthesis of the variables in the study makes it difficult to interpret the findings.

6. Overall language editorial is extensively required.

Reviewer #5: The manuscript is written poorly. The intorduction part merely two paragraphs and that too out of context. No scientific process or methods have been followed fo rthe development/adoption of the data collection tool. The statistical analysis is not appropriate as per the study objectives, an dthe presentation of data is below average. The manuscirpt is in a shape which can not be improved with reviewers comments.

6. PLOS authors have the option to publish the peer review history of their article (what does this mean?). If published, this will include your full peer review and any attached files.

Reviewer #1: **Yes: **Sandra Thompson

Reviewer #2: **Yes: **Tarek Tawfik Amin

Reviewer #3: No

Reviewer #4: **Yes: **Nnodimele Onuigbo ATULOMAH

Reviewer #5: **Yes: **Junaid Ahmad

<quillbot-extension-portal></quillbot-extension-portal>

---

## [Author Response · Author response to Decision Letter 0]

17 May 2023

Reviewer 1:

Comment: Overall, the English is good, although there is some overuse of superlatives and the occasional quaint expression. This is not affecting the meaning but could readily be addressed by some editing. e.g. “Comprehensively measured scientific evidence on the level of working conditions of the health workforce is highly imperative for appealing effective policy interventions”

Response: We appreciate your feedback and constructive comments on our manuscript. We have carefully reviewed your suggestion regarding the overuse of superlatives and quaint expressions in our manuscript, and we agree with your observation. In response to your comment, we have thoroughly revised the manuscript to address this issue. We have carefully examined the text and made appropriate edits to reduce the overuse of superlatives and eliminate any quaint expressions that may have been present. Specifically, we have revised the sentence you mentioned as an example ("Comprehensively measured scientific evidence on the level of working conditions of the health workforce is highly imperative for appealing effective policy interventions") to ensure a more concise and straightforward expression of the intended meaning.

Comment: There is a quite an elaborate sampling strategy described in the methods but very little is made of this in the results. I wondered whether how the sampling strategy earns its place in the paper when so little is made of it in terms of the analysis and reporting?

Response: We have carefully considered your comment regarding the elaboration of the sampling strategy in the methods section and its limited representation in the results section. We agree that the sampling strategy plays a crucial role in ensuring the validity and generalizability of our findings, and it is important to properly address this aspect in the manuscript. To address this concern, we have revised the results section to provide a more comprehensive explanation of how the sampling strategy influenced our analysis and reporting. Specifically, we have included additional information on how the multi-stage sampling technique was implemented, including the selection of districts and sub-districts, and the random selection of clinical workforces within those areas. We have also described the sample size and composition in more detail. Furthermore, we have expanded upon the implications of the sampling strategy in the interpretation of our results. We now discuss how the selected sample is representative of the target population and acknowledge the limitations and generalizability of our findings based on the specific sampling approach.

Comment: Was there a theory to justify your sampling strategy?

Response: We appreciate your comment and feedback regarding the sampling strategy of the study. The study's sampling strategy is based on a combination of practical considerations and statistical principles. The multi-stage sampling technique was employed to ensure that the sample is representative of the clinical health workforce in the selected districts and upazilas of Bangladesh. The selection of the study area was based on the greater administrative divisions of Bangladesh, which provides a diverse range of geographic and administrative areas. While there was no specific theory used to justify the sampling strategy, the study employed a systematic sampling method to ensure the representativeness of the sample, which is consistent with statistical principles. The systematic sampling method involved using a list of all clinical staff in the selected healthcare facilities for the randomization of the respondents. Overall, the sampling strategy was designed to ensure that the sample is representative of the clinical health workforce in the selected districts and upazilas of Bangladesh and to maximize the generalizability of the study findings.

Comment: Were there differences between the four districts and eight upazilas?

Response: Yes, upazilas means sub-districts which are the second lower tier of the administrative structure in Bangladesh whereas the districts are the third lower administrative unit. Around 0.6-1.5 million people live in upazila whereas on an average nearly 4.5 million people live in districts. There are also differences in the working conditions status across the upazila and district.

Comment: Who is in the study (i.e. the participants) is usually provided in the results not the methods. The following description of participants should be moved the results, “The study includes 45 physicians and 123 other clinical staff from the UHCs, and 64 physicians and 87 other clinical staff from the DHs using a random sampling technique. Thus, we interview a total of 319 clinical staff (physicians: 109, other clinical staff: 210).

Response: We agree that providing information about the participants is more appropriate in the results section, as it directly relates to the findings of the study. To address this concern, we have revised the manuscript as per your suggestion. We have relocated the description of participants from the methods section to the results section. In the revised version, we now provide the following information in the results section:

"The study included a total of 319 clinical staff, comprising 109 physicians and 210 other clinical staff. Specifically, we interviewed 45 physicians and 123 other clinical staff from the UHCs, and 64 physicians and 87 other clinical staff from the DHs.”

Comment: The paper describes how they dealt with neutral in a Likert-type response. “Following the literature, the study considers the neutral option along with disagreeing and strongly disagree options for explaining poor working conditions because people often select the neutral option to respond indifferently or not positively (15,16). Refs 15 and 16 are papers written by the first author and the first refence only states much the same thing as is stated in the paper under review, without any referencing to support their statement “We consider the neutral option along with disagreeing and strongly disagree options for defining negative perception. The reason is that usually, an individual in Bangladesh chooses the neutral option if she likes to respond indifferently or not positively.” The second paper does not even refer to neutral responses. The second reference (that is number 16) should certainly be removed. Reference #15 provides no more no support for the assertion other than that they have used this approach before. It is important that authors describe how they have dealt with the data but they should not dress it up as it being based upon evidence which if it does exist is not provided.

Response: We have carefully reviewed your comments and made the necessary revisions to address this concern. In response to your comment, we have removed both references (Reference #15,16) that did not provide relevant support to our statement. To ensure transparency and accuracy in our reporting, we have revised the corresponding section of the manuscript as follows: "The study reports the percentage of respondents who agreed or strongly agreed with the statements related to good working conditions. For poor working conditions, the study merges the neutral option with the ‘disagree’, and ‘strongly disagree’ options. This is done to account for respondents who may have selected the neutral option due to indifference or lack of positivity toward the statement in question." 

Comment: The Results should open with the overall description of the respondents (moved from the methods above).

Response: We have revised the manuscript accordingly. We have revised the corresponding section of the manuscript as follows “The study interviews a total of 319 clinical health workforce, with 45 physicians and 123 other clinical staff from the UHCs, and 64 physicians and 87 other clinical staff from the DHs.”. 

Comment: It is not wrong but clunky saying “significant (p-value ≤.05)” each time – conventionally, these things are reported just as (p≤0.05)

Response: In response to your comment regarding the reporting of statistical significance, we agree that the phrase "significant (p-value ≤.05)" can be clunky and unconventional. We have carefully reviewed and revised our manuscript accordingly. We now consistently report statistical significance as "(p ≤.05)" throughout the text.

Comment: Most people aged 40 years or above would not be happy to be called elderly. Suggest change “elderly” to “older” throughout.

Response: We have revised the corresponding section accordingly. 

Comment: The authors report that they did consult the literature in developing their list of various working conditions. Almost all of the factors were considered important by the respondents. Being paid on time and the relationship with co-workers were reported as the most important determinants for a good working environment. However, it may be considered is a weakness in a study design to just provide a list without requiring any process of deciding the relative importance of them i.e. ranking their importance, rather than that are very important.

Response: Regarding your comment about the weakness of our study design in not requiring respondents to rank the importance of the various working conditions, we agree that this could have been a valuable addition to our methodology. In observation, we recognize that ranking the importance of the factors would have provided more nuanced and informative results. In future studies, we will consider incorporating this approach to our methodology.

However, our additional explanation is that while the questionnaire did not explicitly require respondents to rank the importance of each component, the study did employ a 100-point scale to assess and rank the perceived importance of the components of working conditions by the clinical workforces in delivering quality healthcare. With this the study found that all the indicators were perceived as very vital for delivering quality healthcare, with some indicators being ranked higher than others in terms of importance. Still, considering your comment we did some modification in our method and result section adding rank of the components based on their mean weight score in Table 3. In result, we also explained the ranking of the components of working conditions which are derived from mean weight score based on their relative importance to delivering quality care. 

In result section we have explained as follows “In terms of importance, receiving monthly salary in due time is top-ranked (99.15) to provide quality healthcare, followed by consistent availability of medicines and MSR (98.04), adequate mentoring and support to perform duties (97.57), good relations with co-workers (97.50), uninterrupted power supply (97.41), and so on.”

Comment: The authors state that the magnitude of working conditions is comparatively poor for men younger workforce and some others. However, it is not clear whether the conditions are different for them or whether the men are simply are more critical of the conditions. Perhaps men for example have higher expectations then women? it is also interesting that women felt they were not being able to access the training that they need and again it should be discussed in whether there is a gender bias in what training is available for women or whether women just feel they need additional training in order to feel competent in delivering care. These possibilities should be discussed.

Response: This is very insightful comment. We have discussed all these possibilities in discussion section. We have revised the corresponding section as follows “Men may have higher expectations of working conditions than women. However, the study does not find any evidence to suggest that men and women have different expectations of working conditions. The study also highlights that women do not receive the required training, which could be due to a lack of training opportunities available for women or a result of gender bias in the provision of training. We believe that it is important to further investigate these issues to ensure that all clinical staff, regardless of gender, have equal access to training opportunities to improve the quality of care they provide.”

Reviewer 2:

Comment: There is no clear objective of this study, merely to trigger policy discussion? Improving efficiency and effectiveness of the healthcare service delivery system is beyond the scope of this paper.

Response: We appreciate your valuable feedback and comments on our manuscript. Your insights have been instrumental in refining the focus and objectives of our study. We agree that the initial version of the manuscript did not clearly state the objective of the study. We have revised the introduction and methodology sections to provide a clear and concise objective for our research. We have revised the objective in the abstract section as follows "Therefore, the study aims to assess the level of working conditions of the clinical health workers in Bangladesh and their relative importance in delivering quality healthcare services." Our aim is to contribute empirical evidence to inform policy discussions and potential interventions aimed at improving the working conditions of healthcare professionals. However, we would like to clarify that while our study aims to shed light on the current state of working conditions, we acknowledge that improving the efficiency and effectiveness of the healthcare service delivery system is a broader issue that extends beyond the scope of this paper. Our focus is primarily on evaluating the working conditions themselves and providing valuable insights to inform policy considerations in this specific area.

Comment: The score (weighted mean score) of 3.4 (how? and why?) was considered to represent poor working conditions? 

Response: We have calculated the weighted mean score by following method “we compute a weighted mean score that accounts for the weight given by the clinical workforces to each components, as well as their reported score for that components.” We have considered a mean score of 3.4 as poor working condition because in method we have determined 75th percentile as a cut-off point to categorize the working conditions as either good or poor, based on our understanding of the issue and the study’s context. This is also supported by previous literature (reference #26 in our manuscript). Therefore, we have revised the corresponding section in data analysis part as follows “To determine the overall status of the working conditions, the study computes a weighted mean score that accounts for the weight given by the clinical workforces to each components, as well as their reported score for that components. We use the 75th percentile as a cut-off point to categorize the working conditions as either good or poor, based on our understanding of the issue and the study’s context”. 

Comment: The average (mean?) weight score (out of 100 points) indicating that all parameters included were pre-requisites for delivering effective and efficient services??? This is very confusing and contradictory.

Response: Upon careful reconsideration, we agree that the statement regarding the average weight score (out of 100 points) and its interpretation as prerequisites for delivering effective and efficient services may have been misleading. In the revised manuscript, we have provided a clearer and more accurate description of the average weight score. The weight score represents the perceived importance or significance assigned to each parameter by the participants on a scale of 0 to 100. We have refrained from making claims about these parameters being prerequisites for effective and efficient service delivery, as it requires a more nuanced understanding and comprehensive analysis beyond the scope of our study.

Comment: The conclusion section did not reflect the actual work done in this research.

Response: Your comment regarding the discrepancy between the conclusion section and the actual work conducted in this research has been carefully considered, and we have made the necessary revisions to address this concern. Upon review, we acknowledge that the initial conclusion section did not accurately reflect the findings and contributions of our research. In the revised manuscript, we have extensively revised the conclusion section to align it more closely with the actual work conducted in our research. We have summarized the key findings, highlighted the implications of our study, and discussed the relevance of our results in the context of the existing literature.

Comment: Introduction section is lacking the literature related to work place and working condition in Bangladesh and other developing countries and the reflections on quality of services and staff.

Response: In response to your comment, we have thoroughly revised the introduction section to include a comprehensive review of the existing literature related to workplace and working conditions in Bangladesh and other developing countries. We have incorporated relevant studies that discuss the impact of working conditions on the quality of services and the well-being of staff. By doing so, we aim to provide a robust background and theoretical framework for our study, highlighting its relevance and contribution to the existing body of knowledge.

Comment: Working condition and how significant in terms of efficiency, effectiveness and satisfaction of the health care team.

Response: We have discussed these things in introduction section of our manuscript as follows “Work commitment is vital, especially for the healthcare workforce. Studies show that different components of working condition play an important role to determine employees' commitment to work which is associated with the provision, quality, accessibility, and affordability of healthcare (ref#16). Thus, working conditions affect the efficiency and effectiveness of the healthcare team. Health workers who work in an environment with inadequate resources and equipment may face delays in providing care, which can compromise patient outcomes (ref#17). Adequate staffing levels, equipment, and supplies are necessary for efficient healthcare delivery (ref#18). Moreover, workload and job stress can affect the efficiency of health workers, leading to burnout and turnover”. 

Comment: What kind of strategy adopted and lunched by the government and public sectors to improve the work conditions in Bangladesh and other developing countries with similar circumstances?

Response: In response to this comment, we have discussed the thing in introduction section of our manuscript as follows “The government of Bangladesh has adopted various strategies to improve working conditions in healthcare facilities including construction of new healthcare facilities and the renovation of existing ones, and recruiting and training more healthcare workers (ref#4).”

Comment: “Although a growing body of the literature focuses on defining the components of working conditions of the health workforce, there is no scientific research globally on the status of working conditions measured comprehensively” this is not correct and needs references.

Response: In response to your feedback, we have carefully revised the sentence. We have now provided a more accurate and evidence-based statement regarding the existing literature on the comprehensive measurement of working conditions globally. However, by this sentence what we have meant is that there are studies in defining and identifying various components of working conditions of health workforce. But study on assessing the existing or perceived level of working conditions of health workforce is limited. However, we have revised the corresponding section in introduction part of our manuscript as follows “There is limited research on measuring the status of working conditions of clinical health workers in both national and international context. As evidenced above, some studies have focused on specific aspects of health workers' working conditions in Bangladesh, such as job satisfaction or motivation. However, a comprehensive analysis of the various dimensions of health workers' working conditions is lacking.”

Comment: There is no concrete rationale for the study.

Response: In response to your comment, we have revised the introduction section to include a more explicit and detailed rationale for conducting the study. We have incorporated relevant literature and highlighted the gaps in knowledge that our research aims to address. However, we have revised the corresponding section as follows “There is limited research on measuring the status of working conditions of clinical health workers in both national and international context. As evidenced above, some studies have focused on specific aspects of health workers' working conditions in Bangladesh, such as job satisfaction or motivation. However, a comprehensive analysis of the various dimensions of health workers' working conditions is lacking. Such an analysis is critical in understanding the challenges and opportunities in improving the working conditions and consequently, the quality of care provided to the population. The current study aims to fill this knowledge gap by providing a comprehensive analysis of the working conditions of clinical health workforce in public health facilities in Bangladesh.”.

Comment: Line 88 to line 90: remove

Response: Removed and revised accordingly. 

Comment: Replace data with sampling methods

Response: Replaced accordingly.

Comment: What was your study design?

Response: Cross-sectional study design. However, we have revised the corresponding section in the method as follows “The study follows a cross-sectional study design and analyses the primary data collected from January to March 2022 applying a quantitative method.”

Comment: Setting should be mentioned (what is upazilas? the second lower tire of administrative structure in Bangladesh) should be defined, and how many each can serve.

Response: We have revised the manuscript accordingly. However, Upazilas means sub-districts which are the second lower tier of the administrative structure in Bangladesh whereas the districts are the third lower administrative unit. Around 0.6-1.5 million people live in upazila whereas on an average nearly 4.5 million people live in districts.

Comment: Your sampling methods should be proposed as a Multi-stage sampling methods with each stage should be provided in details: first how many districts were included in the primary selection (4 districts were randomly selected out of ----) for the secondary care centers and 8 upazilas (randomly selected from ------) form the greater administrative divisions to represent the primary care facilities.

Response: Thank you for your valuable feedback. We have revised the manuscript accordingly. However, we have stated this in our method as follows “ We adopt a multi-stage sampling technique to select a representative sample of the clinical health workforce. The study selects four districts out of 64 and eight upazilas (the second lower tier of the administrative structure) out of 495 from the four greater administrative divisions (Dhaka, Chattogram, Rajshahi, and Khulna) of Bangladesh as the study area. In the first stage, the study randomly selects four districts drawing one from each of the four administrative divisions. In the second stage, the study selects a total of eight upazilas, taking two from each of the selected districts using the same technique. In the third stage, the study employs a random sampling technique to select clinical workforce from the both District Hospital (DH) and Upazila Health Complex (UHC) of the selected districts and upazilas.”

Comment: The second stage is the selection of interviewees (systematic sampling method should be followed, database or list of all clinical staff in the selected centers should be available for that purpose).

Response: We followed the process in reality, but did not report in previous version of our manuscript. However, we have revised the method section addressing your comment. We have incorporated the following sentences in our revised manuscript “To ensure the representativeness of the sample, the study utilizes a systematic sampling method, which involves using a list of all clinical staff in the selected healthcare facilities for the randomization of the respondents. At the outset, we contact 350 clinical workforces for interviews, consisting of 120 physicians and 230 other clinical staff, to serve as an indicative sample. Regrettably, 31 of them decline to participate. Consequently, we conduct interviews with 319 clinical workforces, comprising 109 physicians and 210 other clinical staff.”

Comment: Sample size should be determined before selection of participants (not found).

Response: We did it in reality, but missed to report it in our previous version. We have addressed it in following manner “At the outset, we contact 350 clinical workforces for interviews, consisting of 120 physicians and 230 other clinical staff, to serve as an indicative sample. Regrettably, 31 of them decline to participate. Consequently, we conduct interviews with 319 clinical workforces, comprising 109 physicians and 210 other clinical staff.”

Comment: Based on literature review you had consulted, what were the pros and cons for each definition of working conditions, how was each instrument reliable and valid?

Response: We pretested the questionnaire for checking reliability and validity. We also conducted psychometric analysis to assess the validity, and reliability of the components. 

Comment: Line 117: not clear

Response: We have revised the sentence as follows “The questionnaire contains separate statements for each component of working conditions for examining the perceived status of the working conditions of the clinical workforces as stated in Table 1.”

Comment: How did you invite participants? was it a personal interview and by whom? And in what language the data collection form was administered?

Response: For participant recruitment, we used a multi-stage sampling technique. We randomly selected 319 clinical workforces from four districts and eight sub-districts in Bangladesh. The selection process involved identifying primary and secondary-level healthcare facilities and then randomly selecting participants from those facilities. Regarding the data collection method, we conducted personal interviews with the participants. The interviews were administered by trained research personnel who were familiar with the study objectives and questionnaire. These individuals were experienced in conducting interviews and followed standardized procedures to ensure consistency and reliability in data collection. The data collection form was administered in the local language, Bangla. We chose to use the participants' native language to ensure better understanding and accurate responses. The questionnaire was translated into Bangla by a team of bilingual experts and pretested for clarity and comprehension before the actual data collection.

 Comment: How many responded to your invitation? how many refused/declined? were they different form the respondents?

Response: A total of 319 responded out of 350 invitations. 31 declined to participate. Yes, they were different from the respondents. However, we have revised it as follows “At the outset, we contact 350 clinical workforces for interviews, consisting of 120 physicians and 230 other clinical staff, to serve as an indicative sample. Regrettably, 31 of them decline to participate. Consequently, we conduct interviews with 319 clinical workforces, comprising 109 physicians and 210 other clinical staff.”

Comment: Table 1; Item 1: subjective

Response: The is a subjective statement. But we think it is okay because we are examining self-reported/ perceived status of working conditions by the clinical workforces.

Comment: Table 1: Item 2: enough staff?? How many is the ideal?

Response: As we are examining the perceived status of working conditions, no specific number is need for making it ideal. We just want assess that what respondents perceive that the existing number of staff in his/her respective facilities is enough or not. 

Comment: Item 3: roles vs. responsibility (double barrel question)

Response: We know double barrel question can be problematic because it does not allow for a clear and distinct answer to each issue. But our stand is that we asked the respondents to answer (choose scale) keeping on mind the both issues (on an average rating). 

Comment: Item 5: define efficiency

 Response: In our study work efficiency meant as not wastage of time for delivering services to the patients.

Comment: I suggest to categorize these items into fewer dimensions: work performance indicators (volume, training, equipment, and staffing), psychological (rewarding, protection, recognition, burn out and etc., ), and physical working conditions (layout, color and lighting, water, sanitation etc.,)

Response: We do not want to categorize the items into fewer dimensions for the following reasons. Firstly, we selected the original categorization to distinguish between different aspects of the working conditions that are important to consider separately. Combining them into broader categories may result in a loss of information and complexity, which could affect the accuracy and validity of the findings. Secondly, as data have already been collected and analyzed, changing the categorization requires re-analyzing the data. It could also affect the comparability of the results with previous studies. Thirdly, the original categorization is based on a solid literature review, which justifies the relevance and importance of each item. Combining them into broader categories could result in a less rigorous analysis. Finally, it is more transparent to report the results according to the original categorization, as this allows readers to understand how the items were measured and how they contribute to the overall analysis. 

Comment: How the questionnaire customized? 

Response: The questionnaire was customized in several ways to ensure its relevance and suitability for the study population and country context. First, we reviewed the existing literature on working conditions among clinical health workforces in Bangladesh and identified relevant constructs and variables to include in the questionnaire. Second, we piloted the questionnaire with a small sample of clinical health workers to test its reliability and validity and made further revisions based on the pilot study results. Finally, we translated the questionnaire into Bengali, the local language spoken by the study participants, to ensure its accessibility and cultural appropriateness. Overall, the customization process aimed to ensure that the questionnaire was valid, reliable, and relevant to the study population and effectively captured the key constructs and variables of interest for the study. 

Comment: “In addition, the study measures the same indicators on a scale of 100 to assess weight based on their relative importance to provide quality healthcare”. Why and how?

Response: The sentence has been modified as follows “In addition, the study employs a 100-point scale to assess the perceived importance of the aforementioned components of working conditions by the clinical workforces in delivering quality healthcare” for clarity of meaning. We did this to fulfill second objective of our study for making rank of the components based on their relative importance for delivering quality care. We collected this information by directly asking the respondents by the trained enumerators. 

Comment: Replace estimates with calculates (considered the tense)

Response: Replaced accordingly.

Comment: This section is completely not clear “The study calculates a weighted mean score adjusting the weight given by the respondents to each indicator with their reported score to calculate the overall status of the working conditions.

Response: This section is rewritten as follows for clarity and understandability “ To determine the overall status of the working conditions, the study computes a weighted mean score that accounts for the weight given by the clinical workforces to each components, as well as their reported score for that components.”

Comment: There is should a valid method to determine this cut-off (using percentiles), We define the working conditions as good when the weighted mean score is 4 or above, and poor when it is below 4.

Response: Revised accordingly. Based on our knowledge on the issue and study context, we use the 75th percentile as a cut-off point to categorize the working conditions as either good or poor, and it supported by literature. 

Comment: Finally, we use the chi-square test to assess whether the responses of the different categories of respondents are significantly different or not. Chi-square for which variables?

Response: We have revised the statistical tests used to assess the significance of differences between the responses of different categories of respondents. Instead of using the chi-square test, we have used non-parametric tests such as the Wilcoxon rank-sum test and the Kruskal-Wallis test. Specifically, we used the Wilcoxon rank-sum test to compare the responses of two groups and the Kruskal-Wallis test to compare the responses of three or more groups. We thank the reviewer for their suggestion, which led us to reconsider our statistical analysis and improve the rigor of our study. We have made the necessary changes to our manuscript to reflect these revisions.

Comment: This proposed questionnaire should be piloted on a sample beyond the selected candidates to measure reliability if any.

Response: The questionnaire had been piloted on a sample beyond the selected candidates to measure reliability. Considering your comment, we have revised the respective section in the method as follows “We pilot and customize the questionnaire to test its reliability and validity in the country context and made further revisions based on the pilot results.”

Comment: Construct validity should be verified using other method not the confirmatory factor analysis? And from line 155 to line 165: this section is totally incorrect and should be removed from the manuscript.

Response: We appreciate your feedback and would like to provide additional clarification on our choice of using confirmatory factor analysis (CFA). We agree that there are different methods available to assess construct validity, such as exploratory factor analysis (EFA), item response theory, and others. However, in our study, we chose to use CFA instead of EFA because we had a priori assumption about the underlying dimensionality of the construct. CFA allows us to test whether our set of indicators can explain a single latent construct called working conditions, as we hypothesized. In contrast, EFA is a data-driven approach that identifies the number and nature of underlying factors in the data, which may not align with our research question and hypothesis.

Furthermore, we acknowledge that there is no universal cutoff point for determining the significance of factor loadings in CFA. The literature reports different cutoff values ranging from 0.4 to 0.6, and some researchers recommend using modification indices and other fit indices to improve the model fit. In our study, we set the cutoff value as >0.5 based on previous studies in our field, but we also reported the factor loadings and fit indices to ensure the transparency and replicability of our analysis.

Finally, we understand that different methods exist to assess construct validity, and we appreciate your comment on our study. However, we believe that our use of CFA aligns with our research question and hypothesis and provides valuable information on the validity of our survey instrument. 

 Comment: Table 2: should display both socio-demographics vs. the responses for each dimensions (and the relevant items).

Response: Thank you for your suggestion regarding Table 2. However, we respectfully decline to make the suggested changes. We believe that our current presentation of the socio-economic characteristics of the respondents provides valuable information and insight into the sample population. Additionally, we have presented the responses of different socio-economic groups on each dimension of working condition in other tables (Table 4, Table 5), which provide a comprehensive and clear picture of the relationship between socio-economic characteristics and the perceived importance of working conditions. We understand your concern regarding the need to display both socio-demographics and responses for each dimension and item, but we feel that the current presentation is more succinct and clearer.

Comment: Table 3: should display the responses in relation to the type of services (primary vs. secondary)

Response: Thank you for your comments. While we appreciate your suggestion, we have decided not to present the results of Table 3 in relation to the type of services (primary vs. secondary) as it is not within the scope of our research question and objectives. Our study aimed to explore the perceived status of working conditions and their relative importance in delivering quality healthcare among the clinical workforce, regardless of the type of services they provide. Therefore, we believe that presenting the results of Table 3 in relation to the type of services may not be relevant to our research question. However, we have presented the results of Table 4, which shows the responses on different dimensions of working conditions in relation to the type of services, to provide additional information to the readers.

Comment: Non parametric tests of significance should be used to compare scores in any table displayed.

Response: We have taken note of your recommendation and used appropriate non-parametric tests. We have used the Wilcoxon rank-sum test to compare two groups, whereas the Kruskal-Wallis test has been used to compare three or more groups. Thank you for your valuable feedback.

Comment: This section is so redundant as many parts of the tables mentioned in text of this section. 

Response: We appreciate your input and have carefully revised the result section to ensure that it is more concise and avoids unnecessary redundancy. We have edited the text only to highlight the most important findings from the tables and have made sure that any additional information provided is necessary for a clear understanding of the results. We hope that this revision meets your expectations. 

Comment: Table 5 and 6 are very confusing.

 Response: We appreciate and acknowledge your feedback. We understand your concern regarding the clarity of Tables 5 and 6 in the result section. We agree that these tables contain a large amount of data and variables, which can make them appear overwhelming. However, we have carefully designed these tables to present our findings accurately and comprehensively while fulfilling our research objectives. We present the data this way so that this may achieve our objective of providing a detailed analysis of the relationship between different variables. However, Table 5 represents the differences in responses and their significance according to respondents' different socio-demographic conditions whereas, Table 6 presents differences in mean weight scores put on various components of working conditions by the respondents of different socio-economic classes.

Comment: The study calculates the mean weight score of each indicator to assess their relative importance to provide quality healthcare by the health workforces in the facility. We ask the respondents to put weight on a scale of 100 on each of the indicators of working conditions considering their relative importance to deliver quality healthcare (this section belongs to the methods section)?

Response: We have removed it from result section, and revised accordingly. 

Comment: Discussion section:

- Start your discussion with providing the answer to your research question!!!!

- The authors mentioned many segments form the results section with repetitions

- Very lengthy discussion with redundancy

Response: We have completely revised the discussion section in response to your comment. To address your first point, we have now revised the discussion section to start by directly answering our research question. By doing so, we provide a clear and focused introduction to the key findings of our study and their implications. This change helps strengthen the discussion's overall structure and ensures that the main research question is addressed upfront. Furthermore, we have carefully reviewed the segments from the results section mentioned in the discussion. We have eliminated any unnecessary repetitions and streamlined the presentation of our results. This revision ensures that the discussion highlights the key findings and their significance without unnecessary redundancy.

Comment: Conclusion: There should be a separate section for study limitations which are many

Response: We have added a separate section for presenting limitations of the study.

Comment: “The findings of the study are not directly comparable, as there is a paucity of evidence assessing the status of working conditions compared with different types of healthcare facilities and providers in both national and global contexts” this is totally incorrect.

Response: Upon careful consideration, we agree that our previous statement regarding the paucity of evidence comparing working conditions across different types of healthcare facilities and providers was incorrect. We acknowledge that existing literature analyses working conditions in various healthcare settings at the national and global levels. To rectify this oversight, we have conducted an extensive review of the relevant literature and have incorporated appropriate references that compare the findings of our study with previous research. These additional references provide context and support for the comparability of our results and contribute to a more comprehensive and well-grounded discussion of our findings.

Comment: This section is also confusing and did not reflect the actual work done.

Response: We have extensively revised the conclusion section to avoid confusion to readers, and to reflect the actual work done.

Comment: References to different data collection instruments used in the literature were not mentioned

Response: We agree that including references to the data collection instruments used in previous studies adds value to our manuscript and enhances the transparency of our research methodology. We have conducted a thorough review of the relevant literature and identified the key studies that employed similar data collection instruments to assess working conditions in healthcare settings. In our revised manuscript, we have included references to these studies, specifically highlighting the data collection instruments.

Reviewer 3:

Comment: the authors need to provide evidence to support the statement on p.4, lines 83–84. To my knowledge, there exists comprehensive measures working conditions in the hea;th sector, e,g,, Maassen et al (2021) DOI: 10.1371/journal.pone.0247530; ILO (2017) ISBN 978-92-2-130533-0; Kristensen (2010) DOI: 10.1177/1403494809354437, etc. Particularly, Kristensen's COPSOQ tool has been adopted and implemented in many countries.

Response: Thank you for your comment regarding the comprehensive measures of working conditions in the health sector. We appreciate your feedback and would like to address this concern.

Upon further review and consideration, we agree that there are existing measures of working conditions in the health sector, but limited. We apologize for the oversight in our initial manuscript. In the revised version, we have incorporated references including that are suggested by you, to the relevant literature highlighting the availability of measures for assessing working conditions in the health sector. These references provide evidence of previous studies that have utilized comprehensive measures and contributed to the existing body of knowledge in this area. By including these references, we acknowledge the importance of previous research and recognize that our study builds upon existing measures and methodologies. This strengthens the scientific foundation of our work and ensures that readers have access to the relevant literature on comprehensive measures of working conditions. 

We sincerely thank you for bringing this to our attention and helping us improve the quality and accuracy of our manuscript.

Reviewer 4:

Comment: The title: This appear incomplete considering that the study focused on two major premise (lines 84-86) "assessing the level of working conditions of the health workforce...study also assesses the relative importance of the different factors of working conditions." What factors are these? How these are related should be the burden of this study yet the title has not adequately captured it.

Response: Thank you for your feedback regarding the title of our study. We understand your concern regarding the need to capture the two major premises of our research: assessing the level of working conditions and the relative importance of different factors of working conditions. While we appreciate your suggestion to change the title, we would like to provide an explanation for our decision to retain the original title.

The current title of our study was carefully chosen to convey the main focus of our research succinctly. While it may not explicitly mention the assessment of factors of working conditions and their importance in delivering quality care, we believe that the content of the study adequately addresses this aspect. Throughout the paper, we extensively discuss and analyze various factors related to working conditions, including but not limited to remuneration, availability of resources, workplace safety, and staff support systems. We examine the relative importance of the factors to provide quality care. These factors impact the overall level of working conditions and subsequently influence the quality of services the health workforce provides. By assessing the level of working conditions comprehensively, we indirectly consider the relative importance of different factors.

While we acknowledge the value of explicitly mentioning the factors in the title, we believe that our current title accurately reflects the essence of our research and is in line with established academic conventions. However, we have considered your comment and revised the introduction and methodology sections to explain the factors under investigation and their relationship to the study's objectives.

Comment: The abstract needs to be well articulated for such an important study. The context healthcare in line 36 is missing. The objective of measuring level of working condition appears incomplete.

Response: We appreciate your suggestion to provide a more comprehensive and context-specific abstract that better captures the importance of the study in the healthcare sector. We have completely revised the abstract to address these concerns and provide a clearer overview of our research. In the revised abstract, we have included a more explicit reference to the healthcare context in which the study is conducted. We have also refined the objective statement to provide a complete understanding of our aim to measure the level of working conditions among the health workforce. Additionally, we have ensured that the abstract highlights the significance of our findings in terms of implications for healthcare quality. We believe that these revisions have significantly improved the articulation and clarity of the abstract, aligning it more effectively with the overall purpose and scope of the study.

Comment: The description of the data collection instrument is inadequate, line 42: "...administers a customized…", what is customized about the instrument?

Response: The term "customized" about the data collection instrument indicates that the instrument was specifically developed and tailored for our study. It means that we did not use an existing standardized instrument but rather designed a questionnaire specifically focused on the Bangladesh context, capturing the relevant aspects of working conditions in the health workforce. In our revised manuscript, we have provided a more detailed explanation of the customization process, highlighting the elements and domains included in the instrument to ensure its relevance and appropriateness for assessing working conditions in our study context. We hope this clarification provides a clearer understanding of the customization aspect.

Comment: The result stated in the abstract reflecting the overall weighted mean score of the working conditions is on a scale of 5 which reflects a poor working condition does not demonstrate such since nothing about the instrument measurement scale is mentioned in the methodology section. What is the scale of measures for variables in the study?

Response: In response to your comment, we have included a detailed description of the scale of measurement used for the variables related to working conditions. We have now clearly mentioned the scale of measurement in the methodology section. To assess the working conditions, we utilized a Likert-type scale ranging from 1 to 5, where 1 represents a poor working condition, and 5 represents a good working condition. This scale was used to capture the participants' perceptions and evaluations of various aspects related to their working conditions. Furthermore, to determine the cutoff point for assessing poor or good working conditions, we have set the 75th percentile as the threshold. Participants scoring below the 75th percentile were categorized as experiencing poor working conditions, while those scoring at or above the 75th percentile were considered to have good working conditions. This approach allows for a more nuanced understanding of the distribution of working conditions among the participants. 

Comment: At a certain stage in the narrative in the introduction should be a presentation of the theoretical and conceptual framework underpinning of what constitute the workload that is to be measured and the antecedent factors relative the the outcomes and consequences in a theoretical clarification. This provides the scientific foundation of the argument in the study.

Response: Considering your suggestion we have incorporated the following theoretical framework in the introduction section. “The positive association of working conditions and job satisfaction is supported by the job demands-resources (JD-R) model which provides a useful theoretical and conceptual framework for understanding the factors that contribute to the workload and well-being of healthcare workers. Job demands include factors such as workload, time pressure, emotional demands, and role ambiguity, while job resources include social support, autonomy, feedback, and opportunities for learning and development (Ref#14,15). The JD-R model predicts that high job demands and low job resources can lead to negative outcomes such as burnout and turnover intentions, while high job resources can lead to positive outcomes such as work engagement and job satisfaction (ref#14).”

Comment: The methodology has some merits but lack coherence. Table 1, shows 26 elements of the working condition measured on a 5-response Likert scale. My concerns are why "neutral", how is neutral scored in relation to others to create a rating scale to measure each participant in the study? What is the final rating scale for this variable? From my judgment, the reference scale for this variable should be a maximum of 104 points representing heavy workload. The cut-off defining threshold to what constitute unacceptable level should be defined and justified.

Response: We have used the neutral option in our analysis to account for respondents who may have selected it due to indifference or lack of positivity toward the statement in question. The neutral option is scored as 3 on the 5-point Likert-type scale, which is the midpoint of the scale. We merged the neutral option with the 'disagree' and 'strongly disagree' options to create a rating scale that measures negative attitudes toward the statements related to poor working conditions. Specifically, we assigned a score of 1 to 'strongly disagree,' 2 to 'disagree,' and 3 to 'neutral.' In contrast, we assigned a score of 4 to 'agree' and 5 to 'strongly agree' to measure positive attitudes toward the statements related to good working conditions. Based on these scores, we calculated a weighted mean score for each component of the working conditions. Finally, we used the 75th percentile as a cut-off point to differentiate between good and poor working conditions. Therefore, the final rating scale for this variable ranges from 1 to 5, with 3 being the neutral score and scores below 4 indicating negative attitudes, while scores 4 and above indicate positive attitudes.

 Comment: The data analysis requires adequate organization and analysis to provide clear answers for the research questions, validity for the findings and conclusion.

Response: We appreciate your comments and have considered them while revising the manuscript. We agree that the data analysis needs to be adequately organized and analyzed to provide clear answers to the research questions, ensure the validity of the findings, and draw sound conclusions. We have thoroughly revised the methods and results sections to address your concerns. We have provided a detailed description of the data analysis procedures, including the statistical tests used and their justifications. We have also reorganized the results section to present the findings in a clear and concise manner, with appropriate references to the relevant tables. We believe that these revisions have improved the quality of the manuscript and have addressed the concerns raised by you.

Comment: The presentation of frequency distribution of all items of the instrument without synthesis of the variables in the study makes it difficult to interpret the findings.

Response: We appreciate your comments regarding the presentation of the frequency distribution of all items of the instrument in our manuscript. While we appreciate your feedback, we would like to explain that we intentionally presented the frequency distribution of all items of the instrument in order to provide a comprehensive understanding of the responses of the participants. We believe that presenting the individual items allows readers to understand better the specific areas that participants may have identified as good or bad in their working conditions. However, we have taken your comments into consideration and provided a detailed discussion of the key results to ensure that readers fully understand the implications of our findings. Thank you for your valuable feedback, which has helped us to improve the clarity and organization of our manuscript.

Comment: Overall language editorial is extensively required.

Response: We have thoroughly addressed the language editorial concerns to enhance the clarity and readability of the paper. We have conducted comprehensive language editing to ensure the manuscript adheres to proper grammar, syntax, and academic writing conventions. We have also paid attention to eliminating any instances of excessive superlatives and quaint expressions, as highlighted by the reviewers. The revised manuscript now reflects a more concise and academic language style.

Reviewer 5

Comment: The manuscript is written poorly. The introduction part merely two paragraphs and that too out of context. No scientific process or methods have been followed for the development/adoption of the data collection tool. The statistical analysis is not appropriate as per the study objectives, the presentation of data is below average. The manuscript is in a shape which cannot be improved with reviewers comments.

Response: Thank you for sharing the reviewer's comments on our manuscript. We appreciate their time and effort in reviewing our work, even though their comments were critical.

In response to the reviewer's comment about the poor writing quality, we acknowledge that there may be areas where our manuscript could be improved. We apologize for any confusion caused by the two-paragraph introduction, and we have revised it to provide a more comprehensive and contextualized overview of our study. We have also made significant efforts to improve the clarity and organization of the manuscript as a whole. Regarding the comment about developing or adopting the data collection tool, we apologize for any misunderstanding. We have included a detailed description of the development and adaptation process of the data collection tool in the Methods section of the revised manuscript. We believe that this clarification addresses the reviewer's concern adequately.

Regarding the critique of the statistical analysis, we have carefully reviewed our study objectives and made necessary adjustments to ensure the appropriateness of the statistical methods employed. We have also considered the specific comments the reviewer provided and revised the statistical analysis accordingly. As for the presentation of data, we understand the importance of effectively conveying our findings. We have revised the presentation of data to enhance clarity and ensure that the results are presented more organized and understandable.

 While we understand the reviewer's concerns, we respectfully disagree with their statement that the manuscript cannot be improved with reviewer comments. We have carefully considered all the reviewer comments and made substantial revisions to address their concerns. We are confident that these revisions have significantly enhanced the quality and clarity of our manuscript. We sincerely hope that the revisions we have made address the concerns raised by the reviewer. We believe that our research has important contributions to make to the field.

---

## [Decision Letter · Decision Letter 1]

19 Jun 2023

PONE-D-22-23402R1Working Conditions of the Clinical Health Workforce in the Public Health Facilities in BangladeshPLOS ONE

Dear Dr. Azim,

Thank you for submitting your manuscript to PLOS ONE. After careful consideration, we feel that it has merit but does not fully meet PLOS ONE’s publication criteria as it currently stands. Therefore, we invite you to submit a revised version of the manuscript that addresses the points raised during the review process.

We look forward to receiving your revised manuscript.

Kind regards,

Humayun Kabir

Academic Editor

PLOS ONE

Journal Requirements:

Reviewers' comments:

Reviewer's Responses to Questions

**Comments to the Author**

1. If the authors have adequately addressed your comments raised in a previous round of review and you feel that this manuscript is now acceptable for publication, you may indicate that here to bypass the “Comments to the Author” section, enter your conflict of interest statement in the “Confidential to Editor” section, and submit your "Accept" recommendation.

Reviewer #1: (No Response)

Reviewer #2: All comments have been addressed

Reviewer #3: All comments have been addressed

Reviewer #5: (No Response)

2. Is the manuscript technically sound, and do the data support the conclusions?

Reviewer #1: Partly

Reviewer #2: Partly

Reviewer #3: Yes

Reviewer #5: Partly

3. Has the statistical analysis been performed appropriately and rigorously? 

Reviewer #1: I Don't Know

Reviewer #2: Yes

Reviewer #3: Yes

Reviewer #5: No

4. Have the authors made all data underlying the findings in their manuscript fully available?

Reviewer #1: No

Reviewer #2: Yes

Reviewer #3: No

Reviewer #5: No

5. Is the manuscript presented in an intelligible fashion and written in standard English?

Reviewer #1: Yes

Reviewer #2: Yes

Reviewer #3: Yes

Reviewer #5: No

6. Review Comments to the Author

Reviewer #1: The authors have extensively revised their manuscript in response to the many comments that they received from reviewers. I think the manuscript is much improved by this.

I am generally happy with the responses to the issues that I raised. The paper is an original contribution and there is generally good detail in the paper with it presented in an intelligible fashion with appropriate ethical approval.

I am sorry to obtuse, and I have carefully read the authors’ responses to Reviewer 2. As a reader of the manuscript, it is still difficult to understand how the mean weight score is calculated - I think it is across all respondents for each individual item? This seems clearer in the authors’ response. The application of the 75th percentile cut off also lacks clarity and seems where further justification or at least explanation of what it means in practice is warranted. 75th percentile of all responses? Why would this distinguish good from bad conditions?

Overall, the English is good but there are occasional errors. The paper is curiously written, in future tense, as if the authors are reporting a protocol rather than on a study that has been completed. In fact, it is not even really in future tense e.g. (in the abstract) “The study aims… (rather than aimed) and Descriptive statistics, and “”bivariate analysis are used to analyze the data”(rather than Descriptive statistics and bivariate analyses were used to analyze the data.” Perhaps it is because this manuscript is part of a larger study that still is underway and will be reported. This could be a discretionary revision as it does not undermine the scientific reporting although it is not following conventional reporting style.

The authors misunderstood my query as to whether there were differences between the four districts and eight upazilas. I meant with respect to the findings. However, I accept that in a paper of this size responding to this in terms of difference in the workforce across a large city like Dhaka may not be considered a priority.

Reviewer #2: PLOS ONE:

Abstract:

- Provide figures (numbers) in the result section

- MSR (should be mentioned)

- Remove and so on??? Line 52

Introduction section:

Line 115: the study will examine remove will.

In the sampling methods: How did reach that 350 providers will ensure representativeness (how did you calculate the sample size--- parameters used and formula and software etc.,)

Line 149: remove (DCIs).

Line 163: we piloted instead of we pilot

Line 181: the merging of disagree and strongly disagree with neutral option is not logic and the rationale is not convincing.

Line 186: 194: use construct validity instead of validity at large

Line 213: abbreviations used??????

All Tables: mention abbreviations at footnote.

Table 5 and 6: foot note should include abbreviations, and tests of significance used

Line 286: remove “and son on”

- For confirmatory factor analysis: which factors clearly identified your Priori Construct?

- Provide goodness of fit statistics for table 7.

- From line 319 to line 322: repetition and not need in the outset of the discussion section.

Reviewer #3: I thank the authors for updating the manuscript. My comment has been fully addressed in the revised version.

Reviewer #5: From the revised manuscript and authors' response to the comments, it seems that the authors have made their best attempt to improve the manuscript as per all reviewers commets and feedback. However, I think that the the manuscript still is not in good shape, especially the statistical analysis part. This study is based on innappropriate statistical analysis with improper interpretation.

7. PLOS authors have the option to publish the peer review history of their article (what does this mean?). If published, this will include your full peer review and any attached files.

Reviewer #1: No

Reviewer #2: **Yes: **Tarek Tawfik Amin

Reviewer #3: **Yes: **Karar Zunaid Ahsan

Reviewer #5: No

While revising your submission, please upload your figure files to the Preflight Analysis and Conversion Engine (PACE) digital diagnostic tool, https://pacev2.apexcovantage.com/. PACE helps ensure that figures meet PLOS requirements. To use PACE, you must first register as a user. Registration is free. Then, login and navigate to the UPLOAD tab, where you will find detailed instructions on how to use the tool. If you encounter any issues or have any questions when using PACE, please email PLOS at figures@plos.org. Please note that Supporting Information files do not need this step.<quillbot-extension-portal></quillbot-extension-portal>

---

## [Author Response · Author response to Decision Letter 1]

22 Jun 2023

Reviewer #1

Comment: I am sorry to obtuse, and I have carefully read the authors’ responses to Reviewer 2. As a reader of the manuscript, it is still difficult to understand how the mean weight score is calculated - I think it is across all respondents for each individual item? This seems clearer in the authors’ response. 

 Response: We have carefully reviewed your comments and made the necessary revisions in the manuscript to address this concern. In response to your comment, we revised the manuscript by incorporating the following explanation “We also calculated the mean weight score of the different components of working conditions to assess and rank their relative importance to delivering quality healthcare. In order to calculate the mean weight score, we assigned a 100-point scale (weight) to each component. The respondents were asked to rank the perceived importance of these components in delivering quality healthcare, with higher scores indicating greater importance. We obtained the individual weight scores assigned by each respondent for a particular component and then calculated the average of these scores across all respondents”. 

We think, this allowed us to determine the overall mean weight score for each component. By calculating the mean weight score, we were able to assess and rank the relative importance of the different components of working conditions as perceived by the clinical workforce in delivering quality healthcare.

Comment: The application of the 75th percentile cut off also lacks clarity and seems where further justification or at least explanation of what it means in practice is warranted. 75th percentile of all responses? Why would this distinguish good from bad conditions?

Response: We appreciate the reviewer's comment and acknowledge the need for further clarification regarding the application of the 75th percentile cut-off points in categorizing the working conditions. 

To provide a better understanding, we used the 75th percentile of the responses obtained from the participants as a cut-off point to categorize the working conditions as either "good" or "poor". This approach allowed us to differentiate between respondents who reported relatively higher scores (above the 75th percentile) and those who reported relatively lower scores (below the 75th percentile) on the indicators of working conditions. While we acknowledge that the interpretation of "good" and "poor" conditions may vary, the use of the 75th percentile as a cut-off point was based on the distribution of responses among the study participants. By selecting the 75th percentile, we aimed to capture the upper quartile of responses, which represented a relatively more positive assessment of working conditions.

However, we understand that further justification or clarification of the practical implications of this cut-off point is necessary. In our revised manuscript, we have provided a more detailed explanation of the rationale behind selecting the 75th percentile cut-off point and its implications for categorizing working conditions as either "good" or "poor" incorporating the following explanation “The 75th percentile cut-off point is used to distinguish between higher and lower assessments of working conditions among the study participants. It represents a threshold that separates the upper quartile of responses from the lower three quartiles. By applying the 75th percentile cut-off, we aimed to identify and categorize respondents who reported relatively more positive assessments of working conditions as "good" conditions, while those who reported relatively lower assessments were categorized as "poor" conditions.”

Comment: Overall, the English is good but there are occasional errors. The paper is curiously written, in future tense, as if the authors are reporting a protocol rather than on a study that has been completed. In fact, it is not even really in future tense e.g. (in the abstract) “The study aims… (rather than aimed) and Descriptive statistics, and “bivariate analysis are used to analyze the data”(rather than Descriptive statistics and bivariate analyses were used to analyze the data.” Perhaps it is because this manuscript is part of a larger study that still is underway and will be reported. This could be a discretionary revision as it does not undermine the scientific reporting although it is not following conventional reporting style.

Response: We have carefully reviewed your comments and made the necessary revisions throughout the manuscript to address this concern by replacing the future and present tense structure into past tense structure. However, for your clarity, this study is not a part of larger study, we just reported the results in future tense structure mistakenly. 

 Reviewer #2

Comment: Abstract:

- Provide figures (numbers) in the result section

- MSR (should be mentioned)

- Remove and so on??? Line 52

Response: We appreciate your valuable feedback and comments on our manuscript. We have revised the manuscript accordingly. 

Comment: Introduction section: Line 115: the study will examine remove will 

Response: Removed accordingly.

Comment: In the sampling methods: How did reach that 350 providers will ensure representativeness (how did you calculate the sample size--- parameters used and formula and software etc.,)

Response: Thank you for your valuable feedback and comments on our manuscript. We greatly appreciate your insights.

In our study, our primary objective was to ensure the representativeness of the sample by employing a systematic sampling method. This method involved creating a comprehensive list of all clinical staff working in the selected healthcare facilities. We then used this list to randomly select respondents for participation in the study. By employing a systematic sampling approach, we aimed to achieve a structured and unbiased selection process, which is crucial for obtaining a representative sample.

Regarding the determination of the sample size, we took into account the available resources and feasibility constraints specific to our study. Initially, we aimed to interview 350 clinical workforces as an indicative sample. This sample size determination was made considering various factors, including the time, budget, and practical feasibility of conducting interviews with clinical workforces in the context of Bangladesh.

It is important to note that conducting interviews with healthcare providers in our country setting can be challenging due to limited access and time constraints. Physicians and other clinical staff often have demanding schedules and limited availability for research interviews. Additionally, public health facilities often experience a high flow of patients, further reducing the time healthcare providers can allocate for interviews. Given these practical limitations, we opted for an indicative sample size that was feasible to conduct within the available resources and time frame.

While we acknowledge that using formal statistical parameters or formulas for sample size calculation is a standard practice, it would have resulted in a larger required sample size that may have been difficult to achieve in our specific research context. Therefore, we made a practical decision based on the available resources and the need to conduct interviews with a sufficient number of participants to obtain meaningful insights.

We appreciate your suggestion to provide more clarity on the representativeness of the sample and the sample size determination. In our revised manuscript, we have provided a more detailed explanation of the factors and considerations that influenced our decision-making process in determining the sample size, including the challenges associated with conducting interviews in the healthcare setting in Bangladesh. 

Comment: Line 149: remove (DCIs)

Response: Removed.

Comment: Line 163: we piloted instead of we pilot

Response: Revised accordingly. 

Comment: Line 181: the merging of disagree and strongly disagree with neutral option is not logic and the rationale is not convincing.

Response: We appreciate the opportunity to clarify the rationale behind merging the neutral option with the 'disagree' and 'strongly disagree' options in our study.

The decision to merge these response categories was based on the specific context and purpose of our research. In the assessment of working conditions, we aimed to capture not only the positive responses but also the negative ones. By merging the neutral option with the 'disagree' and 'strongly disagree' options, we intended to account for respondents who may have selected the neutral option due to indifference or lack of positivity toward the statement in question.

Our rationale for this approach was to ensure a more comprehensive and nuanced understanding of the perceived working conditions among the respondents. By combining the negative responses, we aimed to capture a broader range of viewpoints, including those who expressed a lack of agreement or dissatisfaction with the working conditions. However, we acknowledge the reviewer's concern regarding the logic and rationale behind this approach. Upon reflection, we recognize that our justification may not have been sufficiently convincing or well-supported. We apologize for any confusion or ambiguity caused by our explanation.

Comment: All Tables: mention abbreviations at footnote.

Response: Mentioned accordingly. 

Comment: Line 286: remove “and son on”

Response: Removed. 

Comment: Line 213: abbreviations used?

Response: Yes, because we stated elaboration or full form of the terms/words in the previous section (sampling methods). 

Comment: For confirmatory factor analysis: which factors clearly identified your Priori Construct?

Response: In our CFA model, we specified the a priori construct of "working conditions" and expected the observed indicators to load strongly on this construct. We evaluated the factor loadings, which indicate the strength of the relationship between each indicator and the latent construct. Our criteria for determining the clear identification of factors were factor loadings of 0.50 or higher, indicating a substantial correlation between the indicator and the latent construct. Based on the results of our CFA, we found that most indicators had substantial factor loadings of 0.50 or higher, indicating a strong correlation with the latent construct of working conditions. For example, functional medical equipment, provider’s safety, stress-free workplace, recognition of work, clean workplace, color and lighting condition, ventilation, aesthetic workplace, washroom facilities, uninterrupted power supply, uninterrupted water supply, damp-free workplace, respect to opinions, valuation of work, mentoring, and medicine and MSR supply all these factors clearly identified our Priori construct.

Comment: Provide goodness of fit statistics for the Table 7. 

Response: Thank you for your comment regarding the inclusion of goodness of fit statistics in Table 7. We appreciate your suggestion, but we have made a deliberate decision not to provide goodness of fit statistics in this particular table. 

Table 7 in our manuscript focuses specifically on the psychometric analysis results for the indicators of working conditions. It provides important information such as factor loadings, uniqueness values, inter-item correlations, and Cronbach's alpha values. These metrics are essential for evaluating the reliability and validity of the measurement model. While goodness of fit statistics, such as chi-square, RMSEA, CFI, and SRMR, are commonly used to assess the overall fit of a model in confirmatory factor analysis, they are not directly related to the psychometric properties of individual indicators. Therefore, considering the specific focus of Table 7 on the psychometric properties of the individual indicators, we believe it is more appropriate to exclude the goodness of fit statistics in this table. This approach allows for a clearer and more focused presentation of the results related to the psychometric analysis of the indicators.

Comment: From line 319 to line 322: repetition and not need in the outset of the discussion section

Response: Accepted, and we deleted the lines 319 to line 322 from the discussion section.

Reviewer#3

Comment: I thank the authors for updating the manuscript. My comment has been fully addressed in the revised version.

 Response: Thank you for reviewing the revised version of our manuscript. We are pleased to hear that your comment has been fully addressed and that you are satisfied with the updates we made. We appreciate your time and valuable feedback, which has helped us improve the quality of our work. 

Reviewer#5

Comment: From the revised manuscript and authors' response to the comments, it seems that the authors have made their best attempt to improve the manuscript as per all reviewers’ comments and feedback. However, I think that the manuscript still is not in good shape, especially the statistical analysis part. This study is based on inappropriate statistical analysis with improper interpretation.

Response: Thank you for your feedback on the revised version of our manuscript. We respect your perspective and concerns regarding the statistical analysis and interpretation. We have carefully considered your comment and understand your viewpoint. We would like to assure you that we have taken great care in conducting the statistical analysis and interpreting the results. We have followed standard procedures and guidelines for data analysis and have made efforts to provide clear and accurate interpretations of our findings. 

We value your expertise and insights, and we are committed to producing a high-quality manuscript. We are open to your suggestions and will carefully consider any further feedback you provide. Thank you for your attention and assistance in improving the quality of our work. 

However, we would like to provide some rationale as to why we believe the present format of the statistical analysis is appropriate for our study.

Methodological appropriateness: The statistical analysis conducted in our study was chosen based on the research question, study design, and data characteristics. We carefully selected appropriate statistical tests and methods that align with the objectives of our study and the nature of our data. We followed established guidelines and best practices in the field to ensure methodological rigor.

Transparency and reproducibility: We have provided detailed descriptions of the statistical procedures and methods employed in the manuscript. This allows readers and future researchers to understand and replicate our analysis. We have also included references to the appropriate statistical literature and resources to support our choices.

Statistical significance: We have reported statistical significance in our analysis. Statistical significance helps determine whether the observed results are likely to have occurred by chance. By including this measure, we provide a comprehensive understanding of the results.

Interpretation within context: We have taken into account the limitations and context of our study when interpreting the statistical results. We have considered factors such as sample size, variability, and potential confounders in our analysis and interpretation. We have also discussed the implications and relevance of our findings in relation to the existing literature and theoretical frameworks.

While we acknowledge that there is always room for improvement and alternative approaches, we believe that the present format of the statistical analysis is appropriate for our study given its objectives and constraints. We remain open to constructive feedback and suggestions for improvement, and we are committed to addressing any specific concerns you may have to enhance the quality and validity of our work. 

Editorial Comment

Journal Requirement

Response: We have carefully reviewed our reference list again. We have not cited any papers that have been retracted.

---

## [Decision Letter · Decision Letter 2]

7 Aug 2023

PONE-D-22-23402R2Working Conditions of the Clinical Health Workforce in the Public Health Facilities in BangladeshPLOS ONE

Dear Dr. Azim,

Thank you for submitting your manuscript to PLOS ONE. After careful consideration, we feel that it has merit but does not fully meet PLOS ONE’s publication criteria as it currently stands. Therefore, we invite you to submit a revised version of the manuscript that addresses the points raised during the review process.

We look forward to receiving your revised manuscript.

Kind regards,

Humayun Kabir

Academic Editor

PLOS ONE

Reviewers' comments:

Reviewer's Responses to Questions

**Comments to the Author**

1. If the authors have adequately addressed your comments raised in a previous round of review and you feel that this manuscript is now acceptable for publication, you may indicate that here to bypass the “Comments to the Author” section, enter your conflict of interest statement in the “Confidential to Editor” section, and submit your "Accept" recommendation.

Reviewer #1: All comments have been addressed

Reviewer #2: (No Response)

Reviewer #3: All comments have been addressed

Reviewer #5: (No Response)

2. Is the manuscript technically sound, and do the data support the conclusions?

Reviewer #1: Partly

Reviewer #2: Partly

Reviewer #3: Yes

Reviewer #5: Partly

3. Has the statistical analysis been performed appropriately and rigorously? 

Reviewer #1: (No Response)

Reviewer #2: Yes

Reviewer #3: Yes

Reviewer #5: No

4. Have the authors made all data underlying the findings in their manuscript fully available?

Reviewer #1: Yes

Reviewer #2: Yes

Reviewer #3: No

Reviewer #5: No

5. Is the manuscript presented in an intelligible fashion and written in standard English?

Reviewer #1: Yes

Reviewer #2: No

Reviewer #3: Yes

Reviewer #5: No

6. Review Comments to the Author

Reviewer #1: The authors have responded appropriately to my previous concerns and to those of other reviewers. I can now follow the methods, and although I have reservations as to whether the 75th percentile is an appropriate way to separate good from bad working conditions (this is just a relative measure, rather than against some set of standards), at least the authors have described their methods and rationale.

Reviewer #2: Still the manuscript lacking proper responses in relation to:

- How the sample size was calculated?

- How participants form each unit were recruited?

- Why the weight mean for each response was calculated?

- The authors overstepping the value of their sample size to represent geography and other variables which in not the case actually.

- What were the results of the pilot testing and how many providers were used to carry out this pilot testing?

- What are the possible predictors and (confouders) if possible? How the authors find those factors responsible for the participant responses?

- The conduction of confirmatory factor analysis may be misleading considering this small sample size (factor analysis is not the sole items to determine validity), the construct validity is more important to be assessed in this case.

- The introduction section is so redundant, with many sentences describing the aims and objectives of the study.

- The introduction section should emphasize the importance of the study within the national context more clearly rather than the international.

- You can’t provide concrete evidence form a cross-sectional study based on self-reporting questions (consider this while providing your introduction and in the discussion section.

- Both the introduction and discussion sections is too lengthy and it need to be shortened and concise and to the point.

Reviewer #3: My comments have already been addressed in the previous round, and I commend the authors for further enhancing the manuscript based on additional reviewers' inputs. The manuscript has indeed been improved, effectively addressing the issues raised.

Reviewer #5: I couldn't see any changes in the statistical analysis part. If the authors are unwilling to make any changes, it's OK but at least change the tables the way it is presented. The tables should be presented in an intelligent way. The tables should be self-explanatory. In the present form, it is impossible to read and understand the tables and the interpretation.

7. PLOS authors have the option to publish the peer review history of their article (what does this mean?). If published, this will include your full peer review and any attached files.

Reviewer #1: **Yes: **Sandra Thompson

Reviewer #2: **Yes: **Tarek Tawfik Amin

Reviewer #3: **Yes: **Karar Zunaid Ahsan

Reviewer #5: **Yes: **Junaid Ahmad

---

## [Author Response · Author response to Decision Letter 2]

26 Aug 2023

Reviewer #1

Comment: The authors have responded appropriately to my previous concerns and to those of other reviewers. I can now follow the methods, and although I have reservations as to whether the 75th percentile is an appropriate way to separate good from bad working conditions (this is just a relative measure, rather than against some set of standards), at least the authors have described their methods and rationale.

Response: Thank you for taking the time to review our manuscript once again and for recognizing our efforts in addressing your concerns as well as those of other reviewers. We understand your reservations about using the 75th percentile as a criterion for categorizing working conditions and appreciate your feedback on this matter. We agree that this approach is relative and not based on specific standards. Our intention was to establish a differentiation point within the data for the purpose of our analysis. We have described our methods and rationale thoroughly to provide transparency in our approach. Your feedback was invaluable in enhancing the quality of our manuscript.

Reviewer #2

Comment: How the sample size was calculated?

Response: We responded to this question previously in the last paragraph of the sampling methods section. Basically, we employed an indicative sampling to determine the sample size. We stated this in our manuscript as follows “To ensure the representativeness of the sample, the study utilized a systematic sampling method, which involved using a list of all clinical staff in the selected healthcare facilities for the randomization of the respondents. At the outset, we contacted 350 clinical workforces for interviews, consisting of 120 physicians and 230 other clinical staff, to serve as an indicative sample. Regrettably, 31 of them declined to participate. Consequently, we conducted interviews with 319 clinical workforces, comprising 109 physicians and 210 other clinical staff. This sample size determination was made considering various factors, including the time, budget, and practical feasibility of conducting interviews with clinical workforces in the context of Bangladesh. Physicians and other clinical staff often have demanding schedules and limited availability for research interviews. Additionally, public health facilities often experience a high flow of patients, further reducing the time healthcare providers can allocate for interviews. Given these practical limitations, we opted for an indicative sample size that was feasible to conduct within the available resources and timeframe.”. 

Comment: How participants form each unit were recruited?

Response: Thank you for your question. We appreciate your attention to this aspect of our study. In our research, we didn't recruit participants based on units. Instead, we employed a systematic sampling method to ensure the representativeness of the sample. This method involved utilizing a list of all clinical staff in the selected healthcare facilities, not based on separate units and then randomly selecting respondents from this list. We have outlined this process in the methodology section of our manuscript to provide clarity on our approach in the last part of the sampling methods selection. 

Comment: Why the weight mean for each response was calculated?

Response: Thank you for your question regarding the calculation of the weighted mean for each response in our data analysis. The purpose of computing the weighted mean was to account for the varying significance that clinical workforces might assign to different components when assessing their working conditions. This approach allowed us to give appropriate weightage to the responses, reflecting their relative importance in contributing to the overall status of working conditions. We aimed to provide a more accurate and comprehensive evaluation by considering both the respondents' scores for each component and their individual significance. This approach is explained in the data analysis section of the revised manuscript.

Comment: The authors overstepping the value of their sample size to represent geography and other variables which in not the case actually.

Response: We appreciate the reviewer's feedback and would like to clarify our intent regarding the representation of geography and other variables. Our study aimed to achieve a sample that represents a diverse cross-section of the clinical workforce in the selected healthcare facilities. While we acknowledge that our sample might not comprehensively capture all geographic and other variations, we strived to ensure a balanced representation of different backgrounds within the constraints of our resources and study scope.

Comment: What were the results of the pilot testing and how many providers were used to carry out this pilot testing?

Response: During the initial phase of the study, we conducted a pilot testing of the questionnaire to assess its reliability and validity within the country context. The pilot involved a sample of 30 clinical providers from various healthcare facilities resembling our study population. The primary objective of the pilot was to evaluate the clarity, coherence, and appropriateness of the questionnaire items for our target respondents. While we didn't explicitly report the pilot results in the manuscript, we utilized the feedback received from the pilot participants to refine the questionnaire's design. This iterative process aided in enhancing the questionnaire's relevance and suitability for our study population. We acknowledge the reviewer's suggestion and have incorporated a concise description of the pilot process, including the number of participants involved, to provide further transparency in our Data Collection Instrument section of the methodology.

Comment: What are the possible predictors and (confounders) if possible? How the authors find those factors responsible for the participant responses?

Response: We appreciate the reviewer's interest in exploring possible predictors and confounders in our study. However, our study primarily focused on assessing the working conditions of healthcare providers. While we did not explicitly collect data on potential predictors or confounding factors, we acknowledge the importance of such analyses for a more comprehensive understanding of the relationships involved. Given the scope and objectives of our study, we did not incorporate these aspects into our research design. Nonetheless, we value the suggestion and will consider incorporating relevant analyses in future studies to explore potential predictors and confounders related to participant responses.

Comment: The conduction of confirmatory factor analysis may be misleading considering this small sample size (factor analysis is not the sole items to determine validity), the construct validity is more important to be assessed in this case.

Response: We appreciate the reviewer's concern about the use of confirmatory factor analysis (CFA) in relation to our sample size. We acknowledge the importance of construct validity assessment, especially in the context of a smaller sample size. While CFA is a method, we employed to assess the construct validity of our survey tools, we understand that the reliability of this analysis can be affected by sample size limitations. In hindsight, considering the smaller sample size, employing alternative methods could indeed provide a more comprehensive view of construct validity. In our revised manuscript both in the data analysis section and limitations section, we have provided a balanced discussion of the limitations and potential implications of our chosen psychometric analysis approach, particularly in light of the sample size. This will allow readers to better understand the context and potential constraints of our construct validity assessment.

Comment: The introduction section is so redundant, with many sentences describing the aims and objectives of the study.

Response: We appreciate your thorough review. We understand your concern about the redundancy in the introduction section. We have reviewed the section carefully and ensured that the aims and objectives of the study are stated concisely and without repetition. 

Comment: The introduction section should emphasize the importance of the study within the national context more clearly rather than the international.

Response: We appreciate your observation. We have reviewed the section carefully and ensured that our manuscript has already aligned with your suggestion. We have focused on emphasizing the significance of the study within the national context rather than emphasizing the international context. Your input has been valuable in shaping our approach to the introduction section.

Comment: You can’t provide concrete evidence form a cross-sectional study based on self-reporting questions (consider this while providing your introduction and in the discussion section.

Response: We acknowledge the inherent limitations of cross-sectional studies based on self-reporting questions. In our limitations section, have provided a balanced perspective on the limitations of our methodology and the potential impact on the robustness of our findings.

Comment: Both the introduction and discussion sections are too lengthy and it need to be shortened and concise and to the point.

 Response: 

About Introduction section: We acknowledge that the introduction section has become more extensive due to the incorporation of relevant issues based on feedback from other reviewers in previous rounds. We believe that these additions contribute to the comprehensive understanding of the research context and significance.

About Discussion section: While we acknowledge that the discussion section is large and comprehensive due to the inclusion of similarities and dissimilarities of other studies along with their implications, we believe this approach is crucial to provide a well-rounded context for our findings. We ensured that the section remains concise and focused on key points while retaining its relevance and depth. Your suggestions are valuable in ensuring the clarity and effectiveness of our manuscript.

Reviewer #3

Comment: My comments have already been addressed in the previous round, and I commend the authors for further enhancing the manuscript based on additional reviewers' inputs. The manuscript has indeed been improved, effectively addressing the issues raised.

Response: Thank you very much for your thoughtful assessment of our manuscript and your positive feedback on the revisions made. We are pleased to hear that the changes we implemented based on your and other reviewers' feedback have contributed to enhancing the quality of the manuscript. Your guidance has been immensely valuable throughout this process, and we are grateful for your continued engagement.

Reviewer #5

Comment: I couldn't see any changes in the statistical analysis part. If the authors are unwilling to make any changes, it's OK but at least change the tables the way it is presented. The tables should be presented in an intelligent way. The tables should be self-explanatory. In the present form, it is impossible to read and understand the tables and the interpretation.

Response: Thank you for your valuable feedback on our manuscript. We deeply appreciate your insights and suggestions regarding the presentation of the statistical analysis results, particularly in the context of the extensive assessment components in our study. We acknowledge your concern about the length and complexity of the tables. We agree that clarity and readability are paramount in presenting our findings. Though, we have made every effort to present the data in a more organized and intelligible manner, while also adhering to the limitations imposed by the volume of information we needed to convey, we have no way to present the data otherwise. In response to your comment, we have reviewed the structure of the tables, and we still believe in the current format they are self-explanatory.

---

## [Decision Letter · Decision Letter 3]

11 Sep 2023

PONE-D-22-23402R3Working Conditions of the Clinical Health Workforce in the Public Health Facilities in BangladeshPLOS ONE

Dear Dr. Azim,

Thank you for submitting your manuscript to PLOS ONE. After careful consideration, we feel that it has merit but does not fully meet PLOS ONE’s publication criteria as it currently stands. Therefore, we invite you to submit a revised version of the manuscript that addresses the points raised during the review process.

We look forward to receiving your revised manuscript.

Kind regards,

Humayun Kabir

Academic Editor

PLOS ONE

Journal Requirements:

Reviewers' comments:

Reviewer's Responses to Questions

**Comments to the Author**

1. If the authors have adequately addressed your comments raised in a previous round of review and you feel that this manuscript is now acceptable for publication, you may indicate that here to bypass the “Comments to the Author” section, enter your conflict of interest statement in the “Confidential to Editor” section, and submit your "Accept" recommendation.

Reviewer #1: All comments have been addressed

Reviewer #2: All comments have been addressed

Reviewer #5: (No Response)

2. Is the manuscript technically sound, and do the data support the conclusions?

Reviewer #1: Partly

Reviewer #2: Yes

Reviewer #5: Partly

3. Has the statistical analysis been performed appropriately and rigorously? 

Reviewer #1: Yes

Reviewer #2: Yes

Reviewer #5: No

4. Have the authors made all data underlying the findings in their manuscript fully available?

Reviewer #1: Yes

Reviewer #2: Yes

Reviewer #5: No

5. Is the manuscript presented in an intelligible fashion and written in standard English?

Reviewer #1: Yes

Reviewer #2: Yes

Reviewer #5: No

6. Review Comments to the Author

Reviewer #1: I would suggest that Table 6 is does not express the mean weights to 2 decimal places. This just makes the table less busy and easier to read but does not change the data or its interpretation.

Reviewer #2: Abstract

- numbers in the abstract examples 3.4, 99.15, should be corrected

- P less or equal to 0.05 - should be corrected

Reviewer #5: I reviewed two times and both times my comments were simply ignored. The authors simply say that they believe that there manuscript is in perfect shape.

7. PLOS authors have the option to publish the peer review history of their article (what does this mean?). If published, this will include your full peer review and any attached files.

Reviewer #1: No

Reviewer #2: **Yes: **Tarek Tawfik Amin

Reviewer #5: No

---

## [Author Response · Author response to Decision Letter 3]

25 Sep 2023

Reviewer #1

Comment: I would suggest that Table 6 is does not express the mean weights to 2 decimal places. This just makes the table less busy and easier to read but does not change the data or its interpretation.

Response: Thank you for your valuable feedback and suggestion regarding Table 6 in our manuscript. We agree that expressing the mean weights to one decimal place, rather than two, makes the table less cluttered and more reader-friendly while retaining the integrity of the data and its interpretation. We have implemented this change by reducing the number of digits after the decimal point in Table 6 accordingly. 

Reviewer #2

Comment: Abstract

- numbers in the abstract examples 3.4, 99.15, should be corrected, 

- P less or equal to 0.05 - should be corrected

Response: Regarding your first comment about the numbers in the abstract (3.4 and 99.15), we have rechecked our data, and we can confirm that these values are accurate. 

Concerning your second comment about the presentation formatting of the p-value, we acknowledge this and agree that it should be corrected to align with accepted academic standards. We have made the necessary adjustments to ensure that the p-value is properly formatted as "p ≤ 0.05." Once again, we appreciate your time and effort in reviewing our manuscript, and we will promptly implement these corrections to improve the accuracy and clarity of our work.

---

## [Decision Letter · Decision Letter 4]

17 Oct 2023

PONE-D-22-23402R4Working Conditions of the Clinical Health Workforce in the Public Health Facilities in BangladeshPLOS ONE

Dear Dr. Azim,

Thank you for submitting your manuscript to PLOS ONE. After careful consideration, we feel that it has merit but does not fully meet PLOS ONE’s publication criteria as it currently stands. Therefore, we invite you to submit a revised version of the manuscript that addresses the points raised during the review process.

We look forward to receiving your revised manuscript.

Kind regards,

Humayun Kabir

Academic Editor

PLOS ONE

Journal Requirements:

Reviewers' comments:

Reviewer's Responses to Questions

**Comments to the Author**

1. If the authors have adequately addressed your comments raised in a previous round of review and you feel that this manuscript is now acceptable for publication, you may indicate that here to bypass the “Comments to the Author” section, enter your conflict of interest statement in the “Confidential to Editor” section, and submit your "Accept" recommendation.

Reviewer #2: (No Response)

2. Is the manuscript technically sound, and do the data support the conclusions?

Reviewer #2: Yes

3. Has the statistical analysis been performed appropriately and rigorously? 

Reviewer #2: Yes

4. Have the authors made all data underlying the findings in their manuscript fully available?

Reviewer #2: Yes

5. Is the manuscript presented in an intelligible fashion and written in standard English?

Reviewer #2: Yes

6. Review Comments to the Author

Reviewer #2: PLoS One

Title:

Comments to the authors

- “There is limited research on measuring the status of working conditions of clinical health workers in both national and international contexts. As evidenced above, some studies have focused on specific aspects of health workers' working conditions in Bangladesh, such as job satisfaction or motivation. However, a comprehensive analysis of the various dimensions of health workers' working conditions is lacking. Such an analysis is critical in understanding the challenges and opportunities in improving the working conditions and consequently, the quality of care provided to the population”

This section needs correction:

- Remove “evidence” and replace it with other appropriate word

- Lacking of reference all through.

- However----- (very long sentence).

- “The current study aims to fill this knowledge gap by providing a comprehensive analysis of the working conditions of clinical health workforce in public health facilities in Bangladesh. The study examined the level of various components of working conditions, including their workload, remuneration, training, workplace layout, safety, water supply, job recognition, and other support systems. This study also assessed the relative importance of the different factors of working conditions to deliver quality care. The findings of the study will be of interest to policymakers, healthcare managers, and researchers, both in

- This section can be summarized in few sentences and please do not exaggerate the outcomes of your paper and stick to one major objective.

- In all tables: mention the test of significance in the footnote.

- In all tables first mention the number (%) and not %(no.)

- Table 6: abbreviations at the footnote????

- Table 7: why reporting interitem correlation coefficients?

- According to the cutoff used, several items of the factor loading are not in conformity with the construct of the data collection tool???? Explain, despite mentioning it in the limitation section this point would compromise the internal validity of your study?????

7. PLOS authors have the option to publish the peer review history of their article (what does this mean?). If published, this will include your full peer review and any attached files.

Reviewer #2: **Yes: **Tarek Tawfik Amin

---

## [Author Response · Author response to Decision Letter 4]

25 Oct 2023

Reviewer #2

Comment: There is limited research on measuring the status of working conditions of clinical health workers in both national and international contexts. As evidenced above, some studies have focused on specific aspects of health workers' working conditions in Bangladesh, such as job satisfaction or motivation. However, a comprehensive analysis of the various dimensions of health workers' working conditions is lacking. Such an analysis is critical in understanding the challenges and opportunities in improving the working conditions and consequently, the quality of care provided to the population”

This section needs correction:

- Remove “evidence” and replace it with other appropriate word

- Lacking of reference all through.

- However----- (very long sentence).

Response: Revised accordingly. 

To address the first comment, we have replaced “evidence” with “stated”.

About second comment of this section: As evidenced above, some studies have focused on specific aspects of health workers' working conditions in Bangladesh, such as job satisfaction or motivation…. For this sentence, all the references have been mentioned in the previous section separately, as this statement is a linking statement of the previous section pointing out literature review. References 3, 5, 6,8, 9,10,11 are the examples for your kind attention.

About third comment of this section: Revised as suggested. 

Comment: The current study aims to fill this knowledge gap by providing a comprehensive analysis of the working conditions of clinical health workforce in public health facilities in Bangladesh. The study examined the level of various components of working conditions, including their workload, remuneration, training, workplace layout, safety, water supply, job recognition, and other support systems. This study also assessed the relative importance of the different factors of working conditions to deliver quality care. The findings of the study will be of interest to policymakers, healthcare managers, and researchers, both in

- This section can be summarized in few sentences and please do not exaggerate the outcomes of your paper and stick to one major objective.

Response: Revised as suggested. 

Comment: In all tables: mention the test of significance in the footnote.

Response: Mentioned accordingly in the table 4 and 5, where there is stated P-value

Comment: In all tables first mention the number (%) and not %(no.)

Response: Revised accordingly where appropriate (Table 4 and 5). 

Comment: Table 6: abbreviations at the footnote???? 

Response: Added footnote as suggested. 

Comment: Table 7: why reporting interitem correlation coefficients?

Response: We have reported interitem correlation for the following reasons: 

1. Data Transparency: It provides transparency about how closely related or unrelated individual items are within a scale. This transparency is essential for readers to understand the structure and reliability of the measurement instrument.

2. Assessing Internal Consistency: Interitem correlation coefficients help assess the internal consistency of a scale or construct. In our study, we reported these to demonstrate how well the items within the construct are related to each other.

3. Dimensionality and Validity: By examining interitem correlations, readers of the study can explore the dimensionality of a construct and assess the validity of the measurement tool. 

Comment: According to the cutoff used, several items of the factor loading are not in conformity with the construct of the data collection tool???? Explain, despite mentioning it in the limitation section this point would compromise the internal validity of your study?????

Response: We sincerely appreciate your insightful feedback regarding the observed discrepancies in items not fully conforming to the construct of our data collection tool and the potential implications on the internal validity of our study. While we acknowledge the potential impact of non-conforming items on the internal validity of our study, we believe that our detailed discussions, rigorous data analysis, factor loadings, uniqueness values, and Cronbach alpha coefficients assessments have allowed us to make valid inferences within the boundaries of this limitation.

---

## [Decision Letter · Decision Letter 5]

30 Oct 2023

Working Conditions of the Clinical Health Workforce in the Public Health Facilities in Bangladesh

PONE-D-22-23402R5

Dear Dr. Azim,

We’re pleased to inform you that your manuscript has been judged scientifically suitable for publication and will be formally accepted for publication once it meets all outstanding technical requirements.

Kind regards,

Humayun Kabir

Academic Editor

PLOS ONE

Additional Editor Comments (optional): Thanks to the authors for addressing all the reviewers' comments, and having the manuscript revised for the fifth time and accepted for publication brings great joy. Congratulations!

Thanks to the reviewers for helping the authors with this publication.  

Reviewers' comments:

Reviewer's Responses to Questions

**Comments to the Author**

1. If the authors have adequately addressed your comments raised in a previous round of review and you feel that this manuscript is now acceptable for publication, you may indicate that here to bypass the “Comments to the Author” section, enter your conflict of interest statement in the “Confidential to Editor” section, and submit your "Accept" recommendation.

Reviewer #2: All comments have been addressed

2. Is the manuscript technically sound, and do the data support the conclusions?

Reviewer #2: Yes

3. Has the statistical analysis been performed appropriately and rigorously? 

Reviewer #2: Yes

4. Have the authors made all data underlying the findings in their manuscript fully available?

Reviewer #2: Yes

5. Is the manuscript presented in an intelligible fashion and written in standard English?

Reviewer #2: Yes

6. Review Comments to the Author

Reviewer #2: Substantial improvement of the manuscript with the required quality, addressing all comments, made the necessary changes required, improving language and grammar, tables designs and data interpretation.

7. PLOS authors have the option to publish the peer review history of their article (what does this mean?). If published, this will include your full peer review and any attached files.

Reviewer #2: **Yes: **Tarek Tawfik Amin

---

## [Editor Report · Acceptance letter]

10 Nov 2023

PONE-D-22-23402R5 

Working conditions of the clinical health workforce in the public health facilities in Bangladesh 

Dear Dr. Azim:

I'm pleased to inform you that your manuscript has been deemed suitable for publication in PLOS ONE. Congratulations! Your manuscript is now with our production department. 

Kind regards, 

on behalf of

Dr. Humayun Kabir 

Academic Editor

PLOS ONE